# HTAC: Hierarchical Task-Aware Composition for Continual Offline Reinforcement Learning

**Qiyang Zhou** [1 2]  **Ruihang Xu** [1]  **Peng Wang** [1]  **Wenjie Lu** [3]  **Xiaochun Cao** [1]  **Naiqiang Tan** [3]  **Li Shen** [1 2]

## Abstract

Continual Offline Reinforcement Learning (CORL) enables building long-term autonomous agents from static datasets. However, it faces heterogeneity in environment dynamics, reward functions, and behavior policies across tasks. Combined with the inherent distribution shift in offline learning, this requires agents to selectively reuse shared knowledge during transfer while isolating task-specific features. The flat knowledge sharing mechanisms employed by existing methods struggle to capture such distinctions, limiting cross-task generalization. To address this, we propose Hierarchical Task-Aware Composition (HTAC), which balances plasticity and stability through dual-level task encoding and soft composition mechanisms. HTAC comprises four modules: (1) a Hierarchical Semantic Task Representation that decomposes tasks into domain-level and task-level embeddings; (2) a Dual-level Expert Network that creates domain and task experts on demand for parameter-efficient knowledge isolation; (3) an Adaptive Knowledge Composition module that integrates historical expert outputs via attention mechanisms for knowledge reuse; (4) Task Adapters that preserve historical routing weights to prevent forgetting. Experiments on Offline Continual World show that HTAC outperforms existing baselines, demonstrating better knowledge reuse and transfer capabilities.

## 1. Introduction

Offline Reinforcement Learning (offline RL) has become a key paradigm in domains such as robotic control (Gürtler et al., 2023) and autonomous driving (Almalioglu et al., 2022), as offline RL learns policies from pre-collected static datasets, avoiding the costs and safety risks associated with online interaction. However, traditional offline RL assumes that all task data can be acquired simultaneously and the environment remains static. In practical applications, data often arrives sequentially—for example, industrial robots must learn different manipulation skills such as grasping, pushing, and placing in succession. Consequently, agents require the capability to continuously accumulate knowledge without accessing historical task environments. Continual Offline Reinforcement Learning (CORL) addresses this challenge by integrating the core principles of offline reinforcement learning (Levine et al., 2020) and continual learning (Masana et al., 2022), enabling agents to continuously accumulate knowledge using only sequential offline data. Unlike standard continual learning, CORL faces a compounded challenge: the inherent distribution shift between behavior policies and learned policies in offline RL (Fujimoto et al., 2019; Kumar et al., 2019) makes model parameters highly sensitive to updates, while heterogeneity across task datasets (Gai et al., 2023; Hu et al., 2024b) further intensifies the stability-plasticity trade-off. Consequently, balancing plasticity for rapid adaptation to new tasks with stability to retain knowledge from previous tasks becomes essential.

Existing CORL methods primarily employ three types of strategies to balance stability and plasticity (Hu et al., 2024e). Regularization-based methods (Kirkpatrick et al., 2017; Aljundi et al., 2018) protect weights important for historical tasks by constraining parameter updates, but under the distribution shift inherent in offline RL, regularization-based methods often fail to effectively balance new and old tasks—either over-constraining parameters and limiting new task learning, or under-constraining parameters and causing catastrophic forgetting. Rehearsal-based methods (Chaudhry et al., 2018b; Wołczyk et al., 2021) alleviate forgetting by storing and replaying historical data, yet rehearsal-based methods incur substantial memory overhead and suffer from distribution mismatch between the replay buffer and the current policy (Gai et al., 2023). Structure-based methods (Mallya & Lazebnik, 2018; Hu et al., 2024e; 2022) allocate independent subnetworks for

---
[1]Shenzhen Campus of Sun Yat-sen University, China [2]Shenzhen Loop Area Institute [3]Didichuxing Co. Ltd. Correspondence to: Li Shen <shenli6@mail.sysu.edu.cn>.

*Proceedings of the 43$^{rd}$ International Conference on Machine Learning*, Seoul, South Korea. PMLR 306, 2026. Copyright 2026 by the author(s).

each task through parameter isolation, effectively preventing forgetting but limiting knowledge transfer. Specifically, structure-based methods typically employ single-layer parameter partitioning without hierarchical decomposition at the representation level, which prevents agents from distinguishing shared domain commonalities from task-specific characteristics and thereby limits cross-task generalization (Hu et al., 2024b;e;a).

To address these limitations, we propose Hierarchical Task-Aware Composition (HTAC). Unlike existing methods that employ flat parameter sharing or isolation, HTAC explicitly models the hierarchical structure of task knowledge by decomposing it into coarse-grained domain knowledge and fine-grained task-specific knowledge, enabling agents to capture multi-level task relationships that facilitate both broad cross-task transfer and precise task-specific adaptation. To realize this hierarchical design, HTAC builds upon Decision Transformer (DT) (Chen et al., 2021) and constructs a dual-level expert architecture where domain experts capture shared knowledge across related tasks while task experts encode task-specific patterns. Meanwhile, HTAC creates experts on demand through a warmup evaluation mechanism, allowing the model to reuse historical knowledge when sufficient and expand its capacity only when necessary. To enable effective knowledge transfer, HTAC introduces an adaptive composition module that uses attention over task embeddings rather than input-dependent routing: domain-level attention aggregates knowledge from previously learned domains weighted by semantic similarity, while task-level attention retrieves the most similar tasks within the same domain. Additionally, HTAC employs Task Adapters to protect historical routing weights, preventing interference with previous task retrieval paths when learning new tasks. Together, these components form a closed-loop process for hierarchical knowledge management: task semantics guide domain assignment and expert routing, warmup evaluation determines whether to reuse or expand expert capacity, and task memory supports future composition. Experiments on the OCW-10 and OCW-20 benchmarks (Hu et al., 2024e) show that HTAC outperforms existing baselines while better balancing plasticity and stability.

In summary, our contributions are threefold:

- We propose HTAC, a hierarchical framework that decomposes task knowledge into domain and task levels, achieving parameter-efficient continual learning through on-demand expert expansion.

- We design a semantic similarity-based composition mechanism using dual-level attention and per-task adapters, enabling forward transfer while preventing catastrophic forgetting.

- Extensive experiments on OCW-10 and OCW-20

demonstrate that HTAC achieves superior average performance and forward transfer compared to existing methods, with negative forgetting rates indicating knowledge enhancement rather than degradation.

## 2. Related Work

### 2.1. Offline Reinforcement Learning

Offline reinforcement learning focuses on learning policies from static datasets without environment interaction (Levine et al., 2020), improving sample efficiency in safety-critical applications (Gottesman et al., 2019; Almalioglu et al., 2022; Gürtler et al., 2023). A key challenge is that learned policies may query out-of-distribution (OOD) actions unseen in the dataset, causing distribution shift and Q-value overestimation (Fujimoto et al., 2019; Kumar et al., 2019). To mitigate these issues, existing approaches can primarily be categorized into three paradigms: (1) Policy constraint methods regularize policies to stay close to behavior policies via KL divergence, MMD, or behavioral cloning (Fujimoto et al., 2019; Fujimoto & Gu, 2021; Nair et al., 2020; Wu et al., 2019); (2) Value regularization learns conservative Q-functions by penalizing OOD actions (Kumar et al., 2020) or avoiding them via expectile regression (Kostrikov et al., 2021); (3) Model-based methods leverage learned dynamics models with uncertainty-aware planning (Yu et al., 2020b; 2021; Kidambi et al., 2020). More recently, sequence modeling approaches reformulate offline RL as conditional generation, using Transformers to model trajectories conditioned on returns (Chen et al., 2021; Janner et al., 2021), eliminating temporal-difference bootstrapping while aligning with supervised learning paradigms (Hu et al., 2024c; Yamagata et al., 2023).

### 2.2. Continual Reinforcement Learning

Continual reinforcement learning (CRL) aims to enable agents to sequentially learn multiple tasks while mitigating catastrophic forgetting (Gai et al., 2023; Abel et al., 2023; Khetarpal et al., 2022). Continual learning methods can be categorized into three main approaches (Masana et al., 2022): regularization-based methods (Kirkpatrick et al., 2017; Aljundi et al., 2018) constrain parameters to preserve prior knowledge; structure-based methods (Mallya & Lazebnik, 2018; Huang et al., 2024) allocate parameter subsets to specific tasks; and rehearsal-based methods (Wołczyk et al., 2021; Chaudhry et al., 2018b) retrain models by merging historical and current task data. These methods have been applied to online RL (Yang et al., 2023; Wołczyk et al., 2021; Zhang et al., 2023). However, adapting these methods to offline RL remains limited (Gai et al., 2023; Huang et al., 2024), giving rise to CORL. CORL addresses scenarios where agents learn from sequentially arriving static datasets without environment interaction (Gai et al., 2023; Hu et al.,

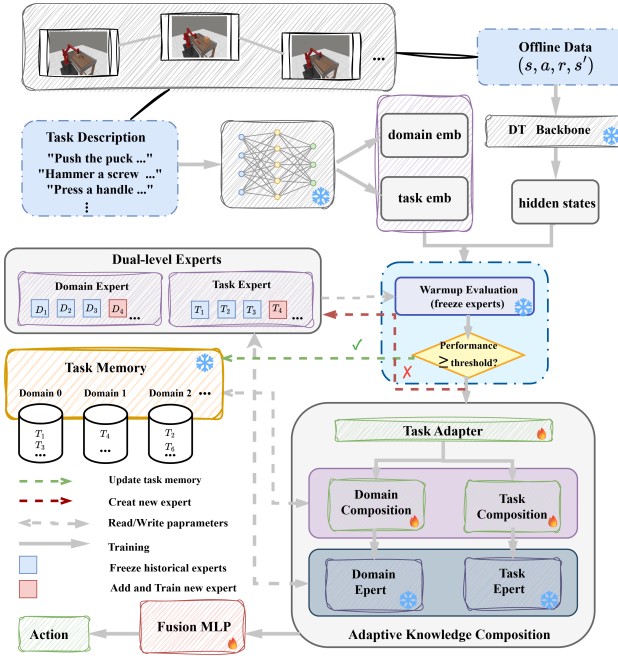

*Figure 1.* Hierarchical Task-Aware Composition architecture. When processing a new task, the system encodes task descriptions hierarchically and evaluates performance using frozen historical experts. If the composed policy meets the performance threshold, it is used directly; otherwise, new domain or task experts are trained and stored in task memory for future composition.

2024b;e). Unlike online CRL, CORL cannot revisit previous environments for data collection (Hu et al., 2024b;a). Existing CORL methods include MH-DT and LoRA-DT (Huang et al., 2024), which extend Decision Transformer for task-specific adaptation; CompoFormer (Hu et al., 2024e), which adaptively composes policies based on task similarity; and OER (Gai et al., 2023), which employs experience selection with dual behavior cloning. Despite these advances, scaling CORL to long task sequences while balancing plasticity and stability remains an open challenge.

## 3. Preliminary

### 3.1. Continual Offline Reinforcement Learning

We consider the task-incremental continual learning setting(Abel et al., 2023; Huang et al., 2024) , where a sequence of $K$ tasks $\mathcal{M}^{(k)}{}_{k=1}^{K}$ arrives sequentially. Each task $\mathcal{M}^{(k)}$ is defined as a Markov Decision Process (MDP) $\langle \mathcal{S}^{(k)}, \mathcal{A}^{(k)}, \mathcal{P}^{(k)}, \mathcal{R}^{(k)}, \gamma^{(k)}, d_0^{(k)} \rangle$, where $\mathcal{S}^{(k)}$ is the state space, $\mathcal{A}^{(k)}$ is the action space, $\mathcal{P}^{(k)} : \mathcal{S}^{(k)} \times \mathcal{A}^{(k)} \to \Delta(\mathcal{S}^{(k)})$ is the transition dynamics, $\mathcal{R}^{(k)} : \mathcal{S}^{(k)} \times \mathcal{A}^{(k)} \to \mathbb{R}$ is the reward function, $\gamma^{(k)} \in [0, 1)$ is the discount factor, and $d_0^{(k)}$ is the initial state distribution.

In the offline setting (Levine et al., 2020), for each task

$\mathcal{M}^{(k)}$, we are provided with a static dataset $\mathcal{D}^{(k)} = \{(s_t^{(k)}, a_t^{(k)}, r_t^{(k)}, s_{t+1}^{(k)})\}$ collected by a behavior policy $\pi_\beta^{(k)}$, without any online environment interaction. The goal of continual offline RL is to learn a policy $\pi^{(k)}$ for each task that maximizes the expected cumulative discounted return $\mathbb{E}_{\pi^{(k)}}[\sum_{t=0}^{\infty}(\gamma^{(k)})^t r_t^{(k)}]$, while maintaining performance on previously learned tasks $\pi^{(i)}{}_{i=1}^{k-1}$ without catastrophic forgetting. Following prior work (Huang et al., 2024; Hu et al., 2024e) , we make the following assumptions: (1) the action space $\mathcal{A}$ is shared across all tasks; (2) task boundaries and identifiers are known to the agent during training; and (3) state spaces have the same dimension across tasks, with variations arising from differences in dynamics $\mathcal{P}$ and reward functions $\mathcal{R}$.

### 3.2. Decision Transformer

Decision Transformer (DT) (Chen et al., 2021) reformulates reinforcement learning as a conditional sequence modeling problem. Given a trajectory of length $T$, DT constructs a sequence of context length $K$ as:

$$\tau_t^{(k)} = \left( \hat{R}_{t-K+1}^{(k)}, s_{t-K+1}^{(k)}, a_{t-K+1}^{(k)}, \ldots, \hat{R}_t^{(k)}, s_t^{(k)}, a_t^{(k)} \right),$$

where $\hat{R}_t^{(k)} = \sum_{i=t}^{T} r_i^{(k)}$ is the return-to-go at timestep $t$. The model is trained to predict actions from state tokens via minimizing the MSE loss:

$$\mathcal{L}_{\text{DT}} = \mathbb{E}_{\tau_t^{(k)} \sim \mathcal{D}^{(k)}} \left[ \frac{1}{K} \sum_{i=t-K+1}^{t} \left\| a_i^{(k)} - \pi_\theta(\tau_t^{(k)})_i \right\|^2 \right],$$

where $\pi_\theta(\tau_t^{(k)})_i$ denotes the $i$-th predicted action.

By casting policy learning as behavior cloning on offline data, DT naturally avoids the distribution shift challenges inherent in offline RL(Levine et al., 2020; Chen et al., 2021). However, in continual learning scenarios, DT's supervised learning paradigm makes it vulnerable to catastrophic forgetting(Huang et al., 2024), as all parameters are updated during training without explicit mechanisms to preserve task-specific knowledge.

## 4. Method

As illustrated in Figure 1, this section first introduces the policy network architecture and training objective, and then presents the design of four core components: Hierarchical Semantic Task Representation, Dual-level Expert Network, Adaptive Knowledge Composition, and Task Adapters.

## 4.1. Policy Network

HTAC adopts the Decision Transformer (Chen et al., 2021) architecture, whose trajectory representation and training objective are introduced in Section 3.2. In the HTAC framework, we extend the standard Decision Transformer to integrate both the hidden state representations from the backbone network and the historical knowledge outputs from the hierarchical composition module:

$$\hat{\mathbf{a}}_t^{(k)} = \mathbf{W}_a^{(k)} \left( \mathbf{h}_L^{(k)} + \sum_{\ell=1}^{L} \mathbf{o}_{\text{composed}}^{(\ell)} \right) + \mathbf{b}_a^{(k)}, \quad (1)$$

where $\mathbf{W}_a^{(k)}$ and $\mathbf{b}_a^{(k)}$ are the learnable parameters of the action prediction head for task $k$, and this action prediction process integrates both the current task representations $\mathbf{h}_L^{(k)}$ and the composed historical representations $\sum_{\ell=1}^{L} \mathbf{o}_{\text{composed}}^{(\ell)}$. HTAC learns by minimizing the MSE loss between predicted and ground-truth actions:

$$\begin{aligned} \pi^{(k)} &= \arg\min_{\theta^{(k)}} \mathcal{L}(\theta^{(k)}) \\ &= \arg\min_{\theta^{(k)}} \mathbb{E}_{\tau \sim \mathcal{D}^{(k)}} \left[ \frac{1}{M} \sum_{m=1}^{M} \|\mathbf{a}_m - \hat{\mathbf{a}}_m\|_2^2 \right], \end{aligned} \quad (2)$$

where $\theta^{(k)}$ denotes the parameters for task $k$, $M$ is the trajectory length, and $\mathcal{D}^{(k)}$ is the offline dataset for task $k$.

## 4.2. Hierarchical Semantic Task Representation Module

To enable knowledge transfer based on semantic similarity, the hierarchical semantic task representation module maps task descriptions to hierarchical semantic representations. The encoder first employs a frozen pre-trained Sentence-BERT model (Reimers & Gurevych, 2019) to encode the textual task description $d^{(k)}$ into a raw semantic embedding $\mathbf{e}_{\text{raw}}^{(k)}$, with fixed parameters to prevent catastrophic forgetting. Subsequently, the encoder performs soft assignment of tasks through $D$ learnable domain prototypes $\{\mathbf{p}_j\}_{j=1}^{D}$:

$$w_{k,j} = \frac{\exp\left(\mathbf{e}_{\text{raw}}^{(k)\top} \mathbf{p}_j / (\tau \|\mathbf{e}_{\text{raw}}^{(k)}\| \|\mathbf{p}_j\|)\right)}{\sum_{m=1}^{D} \exp\left(\mathbf{e}_{\text{raw}}^{(k)\top} \mathbf{p}_m / (\tau \|\mathbf{e}_{\text{raw}}^{(k)}\| \|\mathbf{p}_m\|)\right)}, \quad (3)$$

where $w_{k,j} \in [0,1]$ represents the membership degree of task $k$ to the $j$-th domain, $\mathbf{p}_j$ is the $j$-th domain prototype vector, and $\tau > 0$ is the temperature parameter, where lower temperatures yield sharper assignments. Using these soft assignment weights, we decompose the raw embedding into domain-level and task-level semantic representations for dual-level expert composition. In OCW, task descriptions are structured around manipulation primitives and objects, making textual similarity a useful prior for identifying related skills. HTAC nevertheless treats semantic similarity only as a routing prior: warmup evaluation creates a new task expert when the composed policy fails to reach $\theta$.

## 4.3. Dual-level Expert Network

We design a dual-level expert network with a hierarchical architecture to store knowledge at different granularities. Domain Experts store coarse-grained domain knowledge shared across tasks, with at most one domain expert per domain, ensuring that all tasks within the same domain share the same knowledge base. Task Experts store fine-grained task-specific knowledge for individual tasks. Each expert employs a low-rank decomposition structure to transform the input hidden state $\mathbf{h}$:

$$\mathbf{o} = \mathbf{W}_{\text{down}} \cdot \phi\left(\mathbf{W}_{\text{up}}\mathbf{h} + \mathbf{b}_{\text{up}}\right) + \mathbf{b}_{\text{down}}, \quad (4)$$

where $\mathbf{W}_{\text{up}}$ and $\mathbf{W}_{\text{down}}$ are the up-projection and down-projection matrices, respectively, and $\phi(\cdot)$ is the GELU activation function.

Expert network creation follows an on-demand expansion strategy. When learning each new task, all experts and the composition module are first frozen to evaluate whether existing knowledge is sufficient to solve the current task. If the evaluation performance exceeds a predefined threshold, the existing knowledge is deemed sufficient and no new task expert is created; otherwise, a new task expert is created and trained to store the task-specific knowledge. During the expert training phase, the composition module remains frozen, ensuring that the new expert learns knowledge that is independent of the composition mechanism and can be effectively reused by subsequent tasks.

## 4.4. Adaptive Knowledge Composition Module

To enable knowledge reuse, we design a hierarchical composition module that dynamically composes outputs from historical experts through a semantic similarity-based attention mechanism. This module maintains a task memory bank $\mathcal{M}$ that stores the hierarchical embeddings and expert identifiers of all historical tasks. Domain-level composition leverages semantic similarity at the domain level to produce $\mathbf{o}_{\text{domain}}$, while task-level composition retrieves a subset $\mathcal{S}_k$ of the most semantically similar historical tasks within the same domain and composes their expert outputs $\mathbf{o}_{\text{task}}$ through attention. Finally, the composed output is:

$$\mathbf{h}_{\text{new}}^{(\ell)} = \mathbf{h}^{(\ell)} + \text{MLP}_{\text{fusion}}\left([\mathbf{o}_{\text{domain}}; \mathbf{o}_{\text{task}}]\right), \quad (5)$$

where $[\cdot; \cdot]$ denotes vector concatenation. The expert parameters of historical tasks are frozen after training and are only selectively activated through the attention mechanism, preserving the policy performance on historical tasks during subsequent learning.

## 4.5. Task Adapter Module

Task Adapters enable composition training while preserving the routing patterns of historical tasks. After expert training is completed, the model enters the composition training

*Table 1.* Performance comparison of continual learning methods on the Offline Continual World benchmark, reporting mean and standard deviation across three random seeds. Results show average Performance (P), Forgetting (F), and Forward Transfer (FWT) on OCW10 and OCW20 benchmarks. Higher is better for P and FWT ($\uparrow$), lower is better for F ($\downarrow$).

| Group | Method | OCW10 | | | OCW20 | | |
|---|---|---|---|---|---|---|---|
| | | P ($\uparrow$) | F ($\downarrow$) | FWT ($\uparrow$) | P ($\uparrow$) | F ($\downarrow$) | FWT ($\uparrow$) |
| MT | MTL | $0.75_{\pm 0.04}$ | – | – | $0.80_{\pm 0.02}$ | – | – |
| Reg | L2 | $0.29_{\pm 0.06}$ | $-0.01_{\pm 0.00}$ | $0.03_{\pm 0.05}$ | $0.20_{\pm 0.01}$ | $0.00_{\pm 0.00}$ | $0.01_{\pm 0.01}$ |
| | EWC | $0.16_{\pm 0.02}$ | $0.67_{\pm 0.02}$ | $0.02_{\pm 0.03}$ | $0.12_{\pm 0.02}$ | $0.69_{\pm 0.02}$ | $0.05_{\pm 0.01}$ |
| | MAS | $0.29_{\pm 0.04}$ | $0.47_{\pm 0.07}$ | $0.01_{\pm 0.01}$ | $0.25_{\pm 0.02}$ | $0.44_{\pm 0.02}$ | $0.01_{\pm 0.02}$ |
| | LwF | $0.21_{\pm 0.04}$ | $0.62_{\pm 0.03}$ | $0.06_{\pm 0.04}$ | $0.10_{\pm 0.01}$ | $0.70_{\pm 0.05}$ | $0.06_{\pm 0.03}$ |
| | RWalk | $0.26_{\pm 0.03}$ | $0.01_{\pm 0.01}$ | $-0.02_{\pm 0.01}$ | $0.17_{\pm 0.01}$ | $0.05_{\pm 0.02}$ | $0.00_{\pm 0.01}$ |
| | VCL | $0.14_{\pm 0.03}$ | $0.68_{\pm 0.04}$ | $0.04_{\pm 0.03}$ | $0.06_{\pm 0.00}$ | $0.74_{\pm 0.04}$ | $0.04_{\pm 0.01}$ |
| | Finetuning | $0.11_{\pm 0.03}$ | $0.73_{\pm 0.04}$ | $0.03_{\pm 0.01}$ | $0.08_{\pm 0.00}$ | $0.77_{\pm 0.03}$ | $0.03_{\pm 0.02}$ |
| Struct | LoRA | $0.54_{\pm 0.03}$ | $0.00_{\pm 0.00}$ | $0.01_{\pm 0.00}$ | $0.54_{\pm 0.01}$ | $0.00_{\pm 0.00}$ | $0.02_{\pm 0.02}$ |
| | PackNet | $0.64_{\pm 0.06}$ | $-0.01_{\pm 0.01}$ | $0.03_{\pm 0.01}$ | $0.57_{\pm 0.04}$ | $0.00_{\pm 0.00}$ | $0.05_{\pm 0.02}$ |
| | Grow | $0.60_{\pm 0.06}$ | $-0.01_{\pm 0.01}$ | $-0.01_{\pm 0.01}$ | $0.61_{\pm 0.02}$ | $0.01_{\pm 0.02}$ | $-0.01_{\pm 0.03}$ |
| | Prune | $0.69_{\pm 0.01}$ | $-0.01_{\pm 0.03}$ | $0.00_{\pm 0.00}$ | $0.72_{\pm 0.03}$ | $0.00_{\pm 0.01}$ | $0.03_{\pm 0.02}$ |
| Reh | PM | $0.26_{\pm 0.01}$ | $0.56_{\pm 0.05}$ | $0.04_{\pm 0.03}$ | $0.26_{\pm 0.08}$ | $0.57_{\pm 0.09}$ | $0.11_{\pm 0.02}$ |
| | A-GEM | $0.12_{\pm 0.04}$ | $0.70_{\pm 0.06}$ | $0.02_{\pm 0.01}$ | $0.09_{\pm 0.01}$ | $0.73_{\pm 0.04}$ | $0.06_{\pm 0.02}$ |
| **Ours** | HTAC | $\mathbf{0.72}_{\pm 0.04}$ | $\mathbf{-0.02}_{\pm 0.04}$ | $\mathbf{0.06}_{\pm 0.10}$ | $\mathbf{0.77}_{\pm 0.01}$ | $\mathbf{-0.04}_{\pm 0.02}$ | $\mathbf{0.38}_{\pm 0.02}$ |

phase: all expert parameters are frozen, and only the adapter for the current task is trained to learn the optimal query transformation. Directly updating the shared projection matrices $\mathbf{W}_Q$ and $\mathbf{W}_K$ in the composition module would alter the routing weights of historical tasks, leading to catastrophic forgetting. To address this, task adapters learn independent query transformations for each task while keeping the shared projection matrices frozen:

$$\tilde{\mathbf{e}}^{(k)} = \mathbf{e}^{(k)} + \sigma_k \cdot \mathbf{W}_{\text{up}}^{(k)} \phi \left( \mathbf{W}_{\text{down}}^{(k)} \mathbf{e}^{(k)} \right), \qquad (6)$$

where $\tilde{\mathbf{e}}^{(k)}$ is the task adapter output and $\mathbf{e}^{(k)}$ is the corresponding task embedding $\mathbf{W}_{\text{down}}^{(k)}$ and $\mathbf{W}_{\text{up}}^{(k)}$ are the down-projection and up-projection matrices, respectively, $\phi(\cdot)$ is the GELU activation function, and $\sigma_k$ is a learnable scaling parameter initialized to zero. During training, only the current task's adapter parameters are trainable, while historical adapters remain frozen. After training, the task's hierarchical embedding and expert identifier are added to the task memory $\mathcal{M}$, and all newly created parameters are frozen for subsequent knowledge reuse.

Algorithm 1 summarizes the complete training procedure.

# 5. Experiment

To comprehensively evaluate HTAC on continual offline reinforcement learning, this section presents systematic experiments on the Offline Continual World (OCW) benchmark. We first introduce the experimental benchmark and evaluation metrics, then compare HTAC with representative continual learning methods across three categories: regularization-based, architecture-based, and rehearsal-based, demonstrating its performance on key metrics including average performance, forgetting, and forward transfer. Finally, through ablation studies, we analyze the contribution of core components including the hierarchical expert structure, knowledge composition mechanism, and training strategy to the final performance, revealing the underlying mechanism by which HTAC achieves the stability-plasticity balance.

## 5.1. Benchmarks

To evaluate HTAC, we adopt the Offline Continual World (OCW) benchmark proposed by Hu et al.(Hu et al., 2024e). The OCW benchmark builds upon the Meta-World (Yu et al., 2020a) framework, extending the Continual World (Abel et al., 2023) framework to the offline reinforcement learning setting. The benchmark provides two task sequences: OCW10 with 10 representative robotic manipulation tasks, and OCW20 that repeats OCW10 twice to assess policy transferability upon task revisitation. Each task is equipped with a corresponding offline dataset; detailed descriptions are provided in Appendix B.

## 5.2. Evaluation Metrics

Following standard evaluation protocols in the continual learning literature(Abel et al., 2023; Rolnick et al., 2019; Chaudhry et al., 2018b) , we adopt the following three key metrics to evaluate method performance:

**Average Performance (AP)**: Measures the average success

*Table 2.* Ablation Study of Key HTAC Components. Average performance (P), forgetting (F), and forward transfer (FWT) are reported on the OCW10 benchmark. All results are averaged over 3 random seeds. We observe that each of these components contributes positively to the final performance.

| Experiment | Domain Expert | Task Expert | Adapter | Pretrain | P $\uparrow$ | F $\downarrow$ | FWT $\uparrow$ |
|---|---|---|---|---|---|---|---|
| **Full Model** | ✓ | ✓ | ✓ | ✓ | $\mathbf{0.72}_{\pm 0.04}$ | $\mathbf{-0.02}_{\pm 0.04}$ | $\mathbf{0.06}_{\pm 0.10}$ |
| w/o Domain Expert | × | ✓ | ✓ | ✓ | $0.64_{\pm 0.02}$ | $0.02_{\pm 0.01}$ | $0.02_{\pm 0.01}$ |
| w/o Task Expert | ✓ | × | ✓ | ✓ | $0.39_{\pm 0.02}$ | $-0.01_{\pm 0.00}$ | $0.00_{\pm 0.00}$ |
| w/o Expert | × | × | ✓ | ✓ | $0.17_{\pm 0.03}$ | $-0.01_{\pm 0.02}$ | $0.01_{\pm 0.01}$ |
| w/o Task Adapter | ✓ | ✓ | × | ✓ | $0.49_{\pm 0.03}$ | $0.18_{\pm 0.01}$ | $0.02_{\pm 0.02}$ |
| w/o Composition Pretrain | ✓ | ✓ | ✓ | × | $0.64_{\pm 0.05}$ | $0.00_{\pm 0.02}$ | $0.02_{\pm 0.03}$ |

rate across all tasks at the end of training.

$$AP = \frac{1}{K} \sum_{i=1}^{K} p_i(K \cdot \delta) \quad (7)$$

where $K$ is the total number of tasks, $p_i(t)$ denotes the success rate of task $i$ at time $t$, and $\delta$ is the number of training steps allocated per task.

**Forgetting (F)**: Quantifies the degree of performance degradation on old tasks after learning new tasks.

$$F = \frac{1}{K} \sum_{i=1}^{K} (p_i(i \cdot \delta) - p_i(K \cdot \delta)) \quad (8)$$

It is defined as the average difference between the performance at the end of training and the performance immediately after learning each task.

**Forward Transfer (FWT)**: Measures how learning historical tasks benefits learning new tasks.

$$FWT = \frac{1}{K-1} \sum_{i=2}^{K} (p_i((i-1) \cdot \delta) - p_i(0)) \quad (9)$$

It is defined as the average difference between the performance before encountering task $i$ (i.e., zero-shot testing) and the performance with random initialization (Lopez-Paz & Ranzato, 2017).

### 5.3. Baselines

We compare HTAC against several baselines and state-of-the-art continual RL methods from three categories: **Regularization-based methods** mitigate forgetting by constraining parameter updates based on their importance to previous tasks. We compare against L2, EWC (Kirkpatrick et al., 2017), MAS (Aljundi et al., 2018), LwF (Li & Hoiem, 2017), RWalk (Chaudhry et al., 2018a), VCL (Nguyen et al., 2017), and naive sequential finetuning (Finetuning) as a lower-bound baseline. **Architecture-based methods** avoid task interference through parameter isolation or expansion. We compare against PackNet (Mallya & Lazebnik, 2018),

LoRA (Huang et al., 2024), and two CompoFormer (Hu et al., 2024e) variants (Grow and Prune) that compose historical policies based on task semantic similarity. **Rehearsal-based methods** combat forgetting by storing and replaying historical data. We compare against PM (Wołczyk et al., 2021) and A-GEM (Chaudhry et al., 2018b). Additionally, we report Multi-Task Learning (MTL) (Hu et al., 2024d), which jointly trains on all tasks, as an upper-bound performance reference.

### 5.4. Main Results

Table 1 presents a performance comparison of HTAC and various baseline methods on the OCW10 and OCW20 benchmarks. Overall, HTAC achieves the best performance on both benchmarks, demonstrating that HTAC can accumulate and leverage knowledge in continual learning scenarios while avoiding catastrophic forgetting. Additional experimental analysis is provided in the Appendix E and F.

**Comparison with Regularization-based Methods.** Regularization-based methods perform poorly in continual offline reinforcement learning scenarios. This is because regularization constraints, while limiting parameter changes, cannot distinguish between shared knowledge and task-specific knowledge, hindering the balance between new and old task performance. Although regularization methods exhibit lower forgetting rates compared to the fine-tuning baseline, they still suffer from catastrophic forgetting. In contrast, HTAC achieves lower forgetting rates and stronger forward transfer capability through hierarchical encoding and explicit knowledge isolation, demonstrating that selective knowledge reuse outperforms coarse-grained parameter constraints.

**Comparison with Rehearsal-based Methods.** Rehearsal-based methods also fail to effectively mitigate catastrophic forgetting in offline settings. Even Perfect Memory (PM), which incorporates a large-capacity buffer, only achieves an average performance of 0.26, suggesting that data replay alone is insufficient to retain historical knowledge in complex CORL scenarios. In contrast, HTAC achieves superior

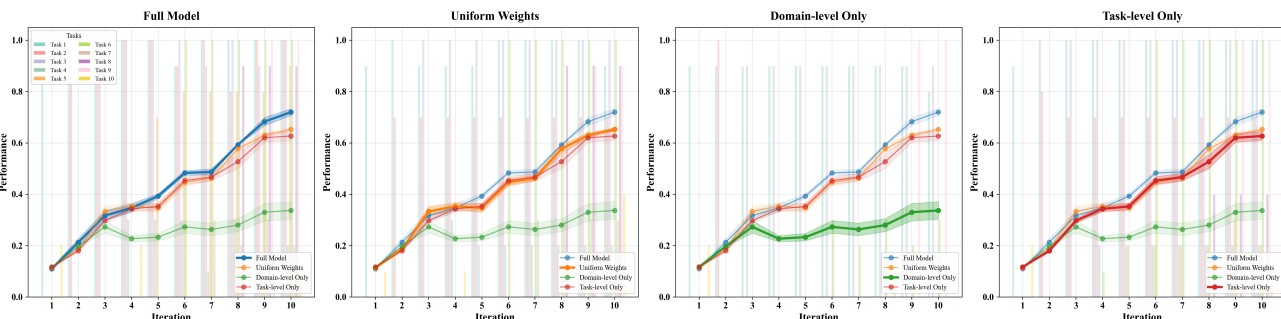

*Figure 2.* Ablation study on composition strategies. Each subplot displays task-level performance and average curves for: (a) Full Model with semantic routing, (b) Uniform Weights, (c) Domain-level Only, and (d) Task-level Only. The x-axis 'Iteration' denotes the sequential training progress, where each iteration corresponds to the completion of learning a new task. All results are averaged over 3 random seeds.

knowledge retention through explicit task-aware modules, eliminating the reliance on historical data storage.

**Comparison with Structure-based Methods.** Structure-based methods outperform the aforementioned categories through parameter isolation strategies. Among them, Prune achieves the best performance by employing binary masking to allocate a sparse sub-network for each task and adaptively composing historical policies via semantic similarity, thereby balancing knowledge sharing with forgetting prevention. However, HTAC surpasses Prune by 4.3% on OCW10 and 5.5% on OCW20. More importantly, HTAC achieves FWT of 0.39 on OCW20, substantially exceeding that of Prune. This indicates that HTAC's adaptive policy composition mechanism effectively promotes knowledge transfer, facilitating both the learning of new tasks and the reuse of historical knowledge.

**Performance on Long Task Sequences.** When the task sequence length increases from 10 to 20, most baseline methods exhibit varying degrees of performance degradation. In contrast, HTAC improves its average performance from 0.72 to 0.77, while FWT increases from 0.06 to 0.39. Since OCW20 repeats the OCW10 sequence, this FWT reflects both transfer to unseen tasks and activation of knowledge acquired during the first pass. As further decomposed in Table 9, HTAC achieves modest positive transfer on the first half and strong knowledge activation on revisited tasks, indicating that semantic routing and frozen expert memory effectively support long-sequence knowledge reuse.

**5.5. Ablation Study**

To analyze the contribution of each component in HTAC, we conducted comprehensive systematic ablation studies on the OCW10 benchmark.

**Hierarchical Architecture Design.** A core design of HTAC is decomposing expert knowledge into two levels: domain-level and task-level. Table 2 presents experimental results under different expert configurations.

As shown in the table, removing Task Experts results in a sharp performance drop, while removing Domain Experts leads to only a modest decline. This difference stems from the functional division between the two types of experts. Task Experts encode task-specific decision details, and their absence directly leads to loss of fine-grained control. Domain Experts store cross-task shared knowledge; while Task Experts can partially compensate for their absence, the increased forgetting indicates that domain-level knowledge is crucial for maintaining historical task memory. When the expert mechanism is completely removed, both performance and FWT drop substantially, validating the necessity of explicit knowledge storage. In summary, Domain Experts maintain cross-task memory and promote forward transfer at low parameter cost, while Task Experts capture task-specific characteristics to ensure decision precision, together forming a shared-specialized hierarchical knowledge system.

**Knowledge Composition Mechanism.** After validating the necessity of the hierarchical architecture, we further analyze the effectiveness of the knowledge composition mechanism.

Figure 2 shows the performance curves and task-level bar charts for each task under four composition strategies. Significant differences can be observed: when only Domain-level composition is retained, the model achieves zero performance on Tasks 2/4/5/7, because domain-level knowledge is too coarse-grained—Tasks 2/4/5/7 within the same pushing domain share similar domain prototypes, but lack task-level experts to distinguish their nuanced differences in trajectory, force, and target position. In contrast, when only Task-level composition is retained, all tasks achieve good learning performance, but forgetting increases markedly, indicating that the absence of domain-level shared knowledge as an anchor weakens the stability of historical tasks. When semantic similarity routing is replaced with uniform weights, performance degrades significantly, demonstrating that uniform weights incorporate irrelevant knowledge, while semantic routing more precisely activates relevant historical task experience. In summary, the dual-level Atten-



*Figure 3.* Final task-wise success rates on the OCW10 benchmark for plasticity evaluation. The gray shaded region represents the single-task training performance, the colored polygons indicate each method's performance after completing the full task sequence. All results are averaged over 3 random seeds.

tion mechanism operates synergistically at two granularities: Domain-level maintains cross-task domain memory, while Task-level ensures fine-grained matching of task-specific characteristics.

**Training Strategy Optimization.** Next, we evaluate the contribution of training strategies to preventing forgetting and improving performance.

As shown in Table 2, after removing the Task Adapter, the gradient updates of the composition module directly interfere with the learned expert knowledge, leading to catastrophic forgetting, as there is no Adapter to provide independent parameter space for each historical task. On the other hand, skipping the Composition Pretraining step that pretrains the composition module on the first 3 tasks leads to a substantial performance drop, indicating that this warm-up phase helps the model establish knowledge retrieval capabilities and provides initialization for subsequent tasks. Task Adapter prevents routing interference through parameter isolation, while Composition Pretraining accelerates knowledge retrieval learning by providing initialization; the two components respectively ensure stability and plasticity.

**Plasticity Analysis.** Finally, we empirically evaluate whether HTAC sacrifices plasticity due to parameter isolation by comparing HTAC with structure-based methods on single-task performance.

HTAC, along with LoRA, Prune, and Grow, belongs to structure-based methods, whose core strategy is to allocate independent parameter subnetworks for each task and freeze them after training, thereby preventing forgetting through parameter isolation. However, this strategy faces the challenge that parameter isolation may limit the model's ability to adapt to new tasks, thereby harming model plasticity. To this end, we use single-task training performance as the upper bound of plasticity and compare the final performance of each method after continual learning against this bound. Figure 3 shows the plasticity radar charts of different methods on the OCW10 benchmark. The closer each method's polygon area is to the gray shaded boundary, the smaller its plasticity loss. As can be observed from the figure: LoRA

exhibits low performance on most tasks, indicating that pure parameter isolation, while preventing forgetting, substantially limits the expressive capacity for new tasks. Although Prune and Grow introduce policy composition mechanisms based on semantic similarity, their composition occurs at the policy output level, relying on complete outputs of previous policies rather than intermediate representations, and thus remain limited in adapting when facing new tasks that differ significantly from historical tasks. In contrast, HTAC closely matches the single-task upper bound on most tasks, and even exceeds the upper bound on Tasks 5 and 8, achieving forward transfer. The key to HTAC's ability to balance stability and plasticity lies in its hierarchical intermediate representation composition mechanism, which performs knowledge composition at the Transformer hidden layers rather than the policy output layer, allowing the model to reuse and adapt historical knowledge at a finer granularity.

## 6. Conclusion

In this work, we investigate the stability-plasticity dilemma in continual offline reinforcement learning. Existing methods employ flat knowledge-sharing mechanisms that fail to differentiate domain-general commonalities from task-specific characteristics, thereby limiting knowledge reuse and transfer. To address this limitation, we propose Hierarchical Task-Aware Composition (HTAC), which decomposes task knowledge into hierarchical representations at domain and task levels through four integrated modules: hierarchical semantic task encoding, on-demand creation of dual-level expert networks, dual-level attention-based adaptive knowledge composition via semantic similarity, and task adapters that preserve historical routing weights. Experiments on the OCW10 and OCW20 benchmarks demonstrate that HTAC outperforms all baseline methods. Ablation studies confirm that domain experts facilitate cross-task knowledge sharing while task experts capture fine-grained specialization; plasticity analysis shows that HTAC maintains near single-task performance across multiple tasks and achieves forward transfer. These results validate that hierarchical knowledge composition based on semantic-aware

routing effectively balances stability and plasticity in long-sequence continual offline reinforcement learning.

**Limitation.** Despite HTAC's superior performance on the OCW benchmarks, the method has room for further improvement. Future work can explore incorporating multimodal task representations that combine task trajectories with textual descriptions, and validating HTAC on broader CORL benchmarks.

## Acknowledgment

This work is supported by NSFC Grant (No. 62576364), GuangDong Basic and Applied Basic Research Foundation (2026B1515020071), Shenzhen Basic Research Project (Natural Science Foundation) Basic Research Key Project (NO. JCYJ20241202124430041), Shenzhen Science and Technology Program (NO.SYSRD20250529113401002), CCF-DiDi GAIA Collaborative Research Funds (NO. CCF-DiDi GAIA 202508).

## Impact Statement

This paper presents work whose goal is to advance the field of Machine Learning. There are many potential societal consequences of our work, none of which we feel must be specifically highlighted here.

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

# Appendix

## A. Algorithm Details

We provide the complete training procedure of HTAC in Algorithm 1. The algorithm consists of three main phases: (1) hierarchical task encoding that decomposes task descriptions into domain-level and task-level embeddings, (2) warmup evaluation that determines whether existing knowledge suffices for the new task, and (3) expert and adapter training when new knowledge acquisition is required.

---

**Algorithm 1** HTAC Training Procedure

---

**Input:** Task sequence $\{(\mathcal{D}^{(k)}, d^{(k)})\}_{k=1}^{K}$, threshold $\theta$, domain prototypes $\{\mathbf{p}_j\}_{j=1}^{D}$, pretrain steps $K_{\text{pre}} = 3$
**Output:** Trained HTAC model with task memory $\mathcal{M}$
Initialize backbone $f_\theta$, composition module, $\mathcal{M} \leftarrow \emptyset$
**for** $k = 1, \ldots, K$ **do**
    $\mathbf{e}_{\text{raw}}^{(k)} \leftarrow \text{S-BERT}(d^{(k)})$; $w_{k,j} \leftarrow \text{softmax}(\mathbf{e}_{\text{raw}}^{(k)\top} \mathbf{p}_j / \tau)$
    $\mathbf{e}_{\text{domain}}^{(k)}, \mathbf{e}_{\text{task}}^{(k)} \leftarrow \text{HierarchicalDecomposition}(\mathbf{e}_{\text{raw}}^{(k)}, \{w_{k,j}\})$
    $d_k^* \leftarrow \arg\max_j w_{k,j}$ {Primary domain assignment}
    **if** $k = 1$ **then**
        Train $f_\theta$ by minimizing $\mathcal{L}(\theta^{(k)})$ (Eq. 2)
        Create DomainExpert$_{d_1^*}$, TaskExpert$_1$; Freeze $f_\theta$
    **else**
        Train $\mathbf{W}_a^{\text{tmp}}$ with frozen experts; $\rho_{\text{warmup}}^{(k)} \leftarrow \text{Eval}(\mathcal{D}^{(k)})$
        **if** $\rho_{\text{warmup}}^{(k)} \geq \theta$ **then**
            Retain $\mathbf{W}_a^{\text{tmp}}$ as $\mathbf{W}_a^{(k)}$
        **else**
            Create DomainExpert$_{d_k^*}$ (if not exists), TaskExpert$_k$
            Train new experts and $\mathbf{W}_a^{(k)}$ with frozen composition
            Freeze experts; Train Adapter$_k = (\mathbf{W}_{\text{down}}^{(k)}, \mathbf{W}_{\text{up}}^{(k)}, \sigma_k)$ {Eq. 8}
            **if** $k \leq K_{\text{pre}}$ **then**
                **Composition Pretraining**: Train $\mathbf{W}_Q, \mathbf{W}_K$ with frozen experts
            **end if**
        **end if**
    **end if**
    Freeze task-$k$ parameters; $\mathcal{M} \leftarrow \mathcal{M} \cup \{(k, \mathbf{e}_{\text{domain}}^{(k)}, \mathbf{e}_{\text{task}}^{(k)}, d_k^*, \text{expert\_ids}_k)\}$
**end for**
Freeze composition module $\mathbf{W}_Q, \mathbf{W}_K$ after $k = K_{\text{pre}}$
**return** Trained model with $\mathcal{M}$

---

## B. Benchmark Details

### B.1. Offline Continual World Benchmark

The Offline Continual World (OCW) benchmark (Hu et al., 2024e) is built on the Meta-World framework (Yu et al., 2020a). It replicates the widely used Continual World (CW) framework (Wołczyk et al., 2021) in the continual offline RL domain by constructing 10 representative manipulation tasks with corresponding offline datasets.

Task Sequence. To increase the benchmark's difficulty, tasks are ranked according to a pre-computed transfer matrix, ensuring significant variation in forward transfer both across the entire sequence and locally (Yang et al., 2023). The OCW10 task sequence is as follows: hammer-v2, push-wall-v2, faucet-close-v2, push-back-v2, stick-pull-v2, handle-press-side-v2, push-v2, shelf-place-v2, window-close-v2, peg-unplug-side-v2

OCW20 Setting. The OCW20 setting repeats the OCW10 task sequence twice to evaluate model performance when revisiting previously learned tasks. This configuration tests both forward knowledge transfer and backward knowledge retention.

Offline Dataset Generation. For each task in the benchmark, the offline dataset was generated by training a Soft Actor-Critic (SAC) (Haarnoja et al., 2018) policy in isolation from scratch until convergence. After convergence, 1 million transitions

were collected from the SAC replay buffer for each task. These transitions comprise samples that were observed during the training process as the policy gradually approaches optimal performance(Yu et al., 2020a).

### B.2. Task Descriptions

Table 3 provides detailed descriptions for each task in the OCW benchmark. These natural language descriptions serve as input to the hierarchical task encoder for computing semantic embeddings.

*Table 3.* Task descriptions in the Offline Continual World benchmark. These natural language descriptions serve as input to the hierarchical task encoder.

| Task | Description |
|------|-------------|
| hammer-v2 | Hammer a screw on the wall |
| push-wall-v2 | Bypass a wall and push a puck to a goal |
| faucet-close-v2 | Rotate the faucet clockwise |
| push-back-v2 | Push the puck to a goal |
| stick-pull-v2 | Grasp a stick and pull a box with the stick |
| handle-press-side-v2 | Press a handle down sideways |
| push-v2 | Push the puck to a goal |
| shelf-place-v2 | Pick and place a puck onto a shelf |
| window-close-v2 | Push and close a window |
| peg-unplug-side-v2 | Unplug a peg sideways |

## C. Hyperparameter Settings

Table 4 summarizes the hyperparameter configurations used in all experiments. We keep the Decision Transformer backbone consistent across all methods for fair comparison.

*Table 4.* Hyperparameter settings for HTAC. The Decision Transformer backbone is kept consistent across all methods for fair comparison.

| Category | Parameter | Value |
|----------|-----------|-------|
| Transformer Architecture | Number of layers | 6 |
| | Number of attention heads | 8 |
| | Hidden dimension | 256 |
| | FFN dimension | 1024 |
| | Dropout | 0.1 |
| | Activation | GELU |
| Training | Batch size | 32 |
| | Learning rate | 1e-4 |
| | Optimizer | Adam |
| | Training steps per task | 5e4 |
| | Context length | 20 |
| HTAC-Specific | Q/K projection dimension | 256 |
| | Performance threshold $\theta$ | 0.8 |
| | Domain expert inner dimension | 1024 |
| | Task expert inner dimension | 512 |

## D. Robustness and Sensitivity Analysis

This section evaluates whether HTAC depends sensitively on key design choices, including the warmup threshold $\theta$, the number of domain prototypes $D$, and the task arrival order.

*Table 5.* Sensitivity analysis of the warmup threshold $\theta$ on OCW20. Results are averaged over 3 random seeds.

| Metric | $\theta = 0.5$ | $\theta = 0.6$ | $\theta = 0.7$ | $\theta = 0.8$ | $\theta = 0.9$ | $\theta = 1.0$ |
|---|---|---|---|---|---|---|
| P | $0.67_{\pm0.05}$ | $0.70_{\pm0.04}$ | $0.71_{\pm0.05}$ | $\mathbf{0.77_{\pm0.01}}$ | $0.74_{\pm0.04}$ | $0.72_{\pm0.03}$ |
| F | $0.01_{\pm0.02}$ | $0.00_{\pm0.02}$ | $-0.00_{\pm0.03}$ | $\mathbf{-0.04_{\pm0.02}}$ | $-0.02_{\pm0.02}$ | $-0.01_{\pm0.03}$ |
| FWT | $0.33_{\pm0.03}$ | $0.35_{\pm0.05}$ | $0.35_{\pm0.05}$ | $\mathbf{0.38_{\pm0.02}}$ | $0.37_{\pm0.03}$ | $0.37_{\pm0.03}$ |

### D.1. Sensitivity to Warmup Threshold $\theta$

The warmup threshold $\theta$ controls whether HTAC reuses historical experts or creates a new task expert. To assess the robustness of this decision rule, we vary $\theta$ on OCW20 while keeping all other settings fixed.

As shown in Table 5, HTAC remains stable across a broad range of $\theta$. A low threshold encourages excessive reuse and may under-allocate task-specific capacity, whereas an overly high threshold triggers more frequent expert creation and weakens reuse. The default value $\theta = 0.8$ provides the best trade-off between performance, forgetting, and forward transfer.

### D.2. Ablation on the Number of Domain Prototypes

We further vary the number of domain prototypes $D$ to evaluate how domain granularity affects hierarchical knowledge sharing. As shown in Table 6, the best performance is obtained at $D = 4$, which matches the natural grouping of OCW manipulation primitives. When $D$ is too small, heterogeneous tasks are forced into the same domain, weakening domain-level sharing. When $D$ is too large, each domain contains too few related tasks, reducing the benefit of hierarchical reuse.

*Table 6.* The number of domain prototypes $D$ on OCW10.

| $D$ | P | F | FWT |
|---|---|---|---|
| 1 | $0.64_{\pm0.02}$ | $-0.00_{\pm0.05}$ | $0.00_{\pm0.01}$ |
| 2 | $0.59_{\pm0.09}$ | $0.01_{\pm0.01}$ | $0.02_{\pm0.01}$ |
| 3 | $0.63_{\pm0.08}$ | $0.03_{\pm0.02}$ | $0.01_{\pm0.02}$ |
| 4 | $\mathbf{0.72_{\pm0.08}}$ | $\mathbf{-0.02_{\pm0.04}}$ | $\mathbf{0.06_{\pm0.10}}$ |
| 5 | $0.59_{\pm0.06}$ | $0.06_{\pm0.02}$ | $0.00_{\pm0.00}$ |
| 6 | $0.67_{\pm0.02}$ | $-0.02_{\pm0.03}$ | $0.00_{\pm0.00}$ |

### D.3. Robustness to Task Arrival Order

To evaluate whether HTAC depends on a specific task curriculum, we test four OCW10 task sequences. The Table 8 show that HTAC consistently outperforms representative baselines across different orders.

*Table 7.* Task sequences used for order robustness evaluation on OCW10.

| Order | Task Sequence |
|---|---|
| 0 | hammer-v2, push-wall-v2, faucet-close-v2, push-back-v2, stick-pull-v2, handle-press-side-v2, push-v2, shelf-place-v2, window-close-v2, peg-unplug-side-v2 |
| 1 | push-v2, window-close-v2, peg-unplug-side-v2, shelf-place-v2, handle-press-side-v2, push-back-v2, hammer-v2, stick-pull-v2, push-wall-v2, faucet-close-v2 |
| 2 | handle-press-side-v2, peg-unplug-side-v2, push-back-v2, stick-pull-v2, push-v2, shelf-place-v2, faucet-close-v2, window-close-v2, push-wall-v2, hammer-v2 |
| 3 | push-wall-v2, handle-press-side-v2, push-v2, hammer-v2, peg-unplug-side-v2, stick-pull-v2, shelf-place-v2, faucet-close-v2, window-close-v2, push-back-v2 |

HTAC achieves the best average performance under all four task orders and shows small variation across orders. This suggests that its performance does not rely on a particular curriculum, but instead comes from the hierarchical expert structure and semantic reuse mechanism.

## E. Detailed Per-Task Performance Analysis

This section investigates the stability-plasticity trade-off in Continual Offline Reinforcement Learning (CORL). Using the OCW10 dataset, we analyze the performance of various methods by comparing their per-task scores against the single-task training upper bound.

**Experimental Setup.** We employ Single Task Training as the performance upper bound. In this setting, a model is trained

*Table 8.* Robustness to different task arrival orders on OCW10. Results are averaged over 3 random seeds.

| Method | Order0 | Order1 | Order2 | Order3 | Avg. |
|---|---|---|---|---|---|
| L2 | $0.29_{\pm 0.06}$ | $0.20_{\pm 0.03}$ | $0.23_{\pm 0.05}$ | $0.29_{\pm 0.05}$ | 0.25 |
| EWC | $0.16_{\pm 0.02}$ | $0.12_{\pm 0.02}$ | $0.11_{\pm 0.02}$ | $0.15_{\pm 0.03}$ | 0.13 |
| MAS | $0.29_{\pm 0.04}$ | $0.17_{\pm 0.01}$ | $0.16_{\pm 0.06}$ | $0.20_{\pm 0.04}$ | 0.21 |
| LoRA | $0.54_{\pm 0.03}$ | $0.47_{\pm 0.02}$ | $0.35_{\pm 0.03}$ | $0.43_{\pm 0.05}$ | 0.45 |
| PackNet | $0.64_{\pm 0.06}$ | $0.67_{\pm 0.01}$ | $0.65_{\pm 0.03}$ | $0.65_{\pm 0.04}$ | 0.65 |
| Grow | $0.60_{\pm 0.06}$ | $0.54_{\pm 0.05}$ | $0.43_{\pm 0.01}$ | $0.51_{\pm 0.05}$ | 0.52 |
| PM | $0.26_{\pm 0.01}$ | $0.26_{\pm 0.10}$ | $0.25_{\pm 0.03}$ | $0.27_{\pm 0.01}$ | 0.26 |
| HTAC | $\mathbf{0.72_{\pm 0.04}}$ | $\mathbf{0.72_{\pm 0.02}}$ | $\mathbf{0.70_{\pm 0.03}}$ | $\mathbf{0.69_{\pm 0.03}}$ | **0.71** |

to convergence on a specific task in isolation, free from interference by other tasks, thereby reflecting the model's theoretical optimal performance for that task. By comparing the final performance of various continual learning methods after a sequence of 10 tasks against this benchmark, we simultaneously evaluate their stability (the ability to retain performance on historical tasks) and plasticity (the capacity to effectively learn new tasks).

Figure 4illustrates the radar charts depicting the per-task performance of all methods on the OCW10 benchmark. In each subplot, the shaded region represents the method's final performance across all tasks, with a larger polygon area indicating superior overall performance. Drawing on this analysis, we discuss the performance of different method categories below:

**Regularization and Rehearsal-based Methods.** Regularization methods (L2, EWC, MAS, LwF, VCL, RWalk) and rehearsal methods (AGEM, PM) mitigate catastrophic forgetting by incorporating regularization terms or gradient constraints into the loss function. Nevertheless, these methods still suffer from forgetting during training, particularly for earlier tasks in the sequence. As shown in the figure, these methods exhibit performance significantly below the single-task upper bound on most tasks, achieving relatively better performance only on tasks at the end of the sequence. While increasing regularization strength can better prevent forgetting, it impairs the capacity to learn new tasks, leading to a pronounced stability-plasticity trade-off. This issue is particularly acute in offline reinforcement learning, where policy optimization exhibits substantially higher sensitivity to parameter changes compared to supervised learning in classification tasks.

**Finetuning Method.** The finetuning method employs no anti-forgetting mechanism and simply finetunes the previous model on each new task. As shown in the figure, this method's performance on all historical tasks degrades to near-zero, with only the most recently trained task maintaining high performance. This extreme catastrophic forgetting demonstrates that effective continual learning is infeasible without explicit stability preservation mechanisms.

**Structure-based Methods.** Structure-based methods (LoRA, PackNet/Prune, Grow) maintain stability by freezing task-specific parameters. However, as training progresses, the capacity of these methods to learn new tasks gradually declines due to the reduction in available parameters and cross-task interference arising from frozen parameters. Specifically: LoRA is constrained by its low-rank approximation, limiting its capacity to fully capture feature representations of complex tasks and resulting in degraded performance on later tasks; PackNet/Prune, while able to free up parameter space for new tasks through network pruning, encounters progressively fewer parameters available for allocation as the task sequence grows; Grow, although introducing a policy composition mechanism based on semantic similarity to enhance plasticity, performs composition at the policy output level rather than at intermediate representations. Consequently, Grow relies on complete outputs from previous policies and struggles to effectively adapt when confronting new tasks that differ substantially from historical ones.

**HTAC.** In contrast, HTAC closely approaches the single-task upper bound on most tasks while maintaining strong stability. Notably, on Task 5 and Task 8, HTAC's performance exceeds the single-task upper bound, achieving positive transfer. This demonstrates that HTAC not only effectively prevents forgetting but also leverages historical task knowledge to facilitate new task learning. HTAC achieves this favorable balance between stability and plasticity through its hierarchical intermediate representation composition mechanism. Unlike methods such as Grow that perform composition at the policy output level, HTAC conducts knowledge composition within the Transformer's hidden layers. This design reduces cross-task interference and enhances plasticity through the following mechanisms: (1) Intermediate representations contain richer semantic information, facilitating fine-grained knowledge reuse and adaptation; (2) The attention mechanism adaptively selects historical knowledge relevant to the current task, avoiding negative transfer from irrelevant task knowledge; (3) The hierarchical domain-task two-tier composition structure simultaneously captures coarse-grained domain commonalities and fine-grained task-specific characteristics, enabling new tasks to more effectively leverage existing knowledge.

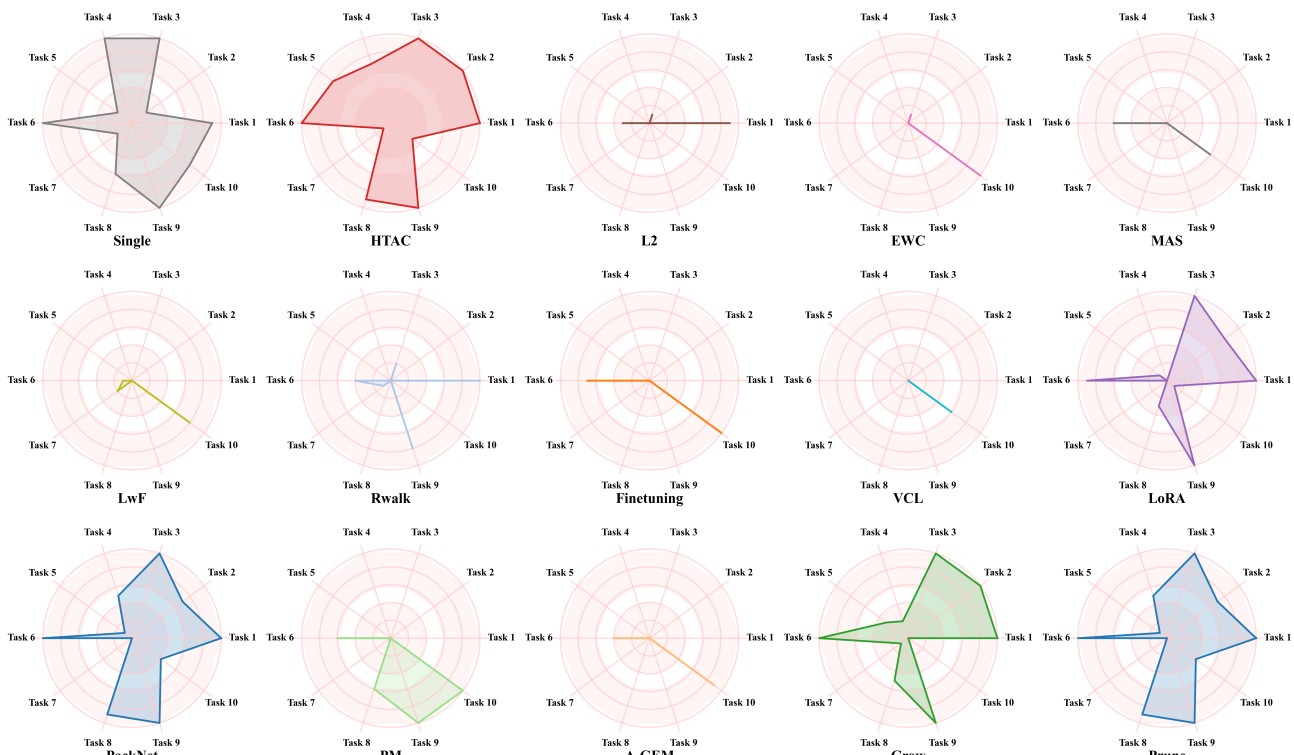

*Figure 4.* Final task-wise success rates on the OCW10 benchmark for plasticity evaluation. The gray shaded region represents the single-task training performance, the colored polygons indicate each method's performance after completing the full task sequence.All results are averaged over 3 random seeds.

In summary, the per-task performance analysis reveals distinct differences among methods in navigating the stability-plasticity trade-off: regularization and rehearsal methods struggle to balance the two objectives; structure-based methods maintain stability at the cost of plasticity; whereas HTAC, through its hierarchical intermediate representation composition mechanism, effectively mitigates cross-task interference, maintains stability while enhancing the capacity to learn new tasks, and achieves positive transfer on certain tasks.

## F. Comprehensive Analysis on OCW20 Benchmark

This section analyzes the performance of HTAC and baseline methods within the OCW20 long-sequence task scenario. Constructed by repeating the OCW10 task sequence twice, OCW20 creates a 20-task continual learning process that evaluates the capacity for retaining old tasks and learning new ones, while specifically enabling the assessment of knowledge reuse and enhancement during task revisitation.

### F.1. Overall Performance Comparison and Learning Dynamics

Figure 5 illustrates the overall performance evolution of various methods on the OCW20 benchmark, where the x-axis represents training epochs and the y-axis denotes the average performance across all learned tasks. Several key observations can be made:

(1) Continuous Performance Improvement and Knowledge Accumulation in HTAC.HTAC exhibits performance growth throughout the 20-task sequence, achieving a final average performance of 0.77 and outperforming all baselines. Notably, during tasks 11–20, the revisitation phase, the slope of the HTAC performance curve increases, indicating effective identification and reuse of prior knowledge. This accumulation effect is twofold: first, for revisited tasks, HTAC activates historical experts via semantic similarity matching, bypassing the need for learning from scratch; second, shared domain-level experts enable subsequent tasks within the same domain to leverage the established knowledge foundation. This is quantitatively supported by the Forward Transfer metric in Table 1, where HTAC achieves an FWT of 0.39±0.04 on OCW20.

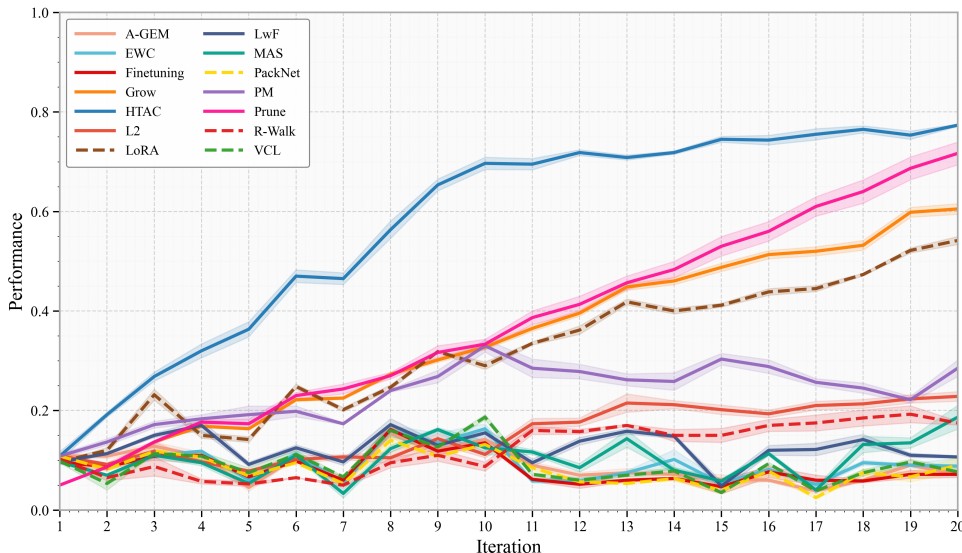

*Figure 5.* Method performance comparison across 20 tasks on OCW20 benchmark. The curves show accumulated average performance throughout the continual learning process. All results are averaged over 3 random seeds.

(2) Performance and Limitations of Structured Methods. Structured methods (Prune, Grow, LoRA, PackNet) generally outperform other baselines. Prune performs best among them, mitigating forgetting via independent sub-networks and semantic policy composition. However, a performance gap remains between Prune and HTAC, as Prune composes knowledge at the policy output layer, lacking fine-grained intermediate reuse. LoRA exhibits performance saturation, with nearly identical results on OCW10 and OCW20 due to low-rank constraints. PackNet shows performance degradation in longer sequences, attributed to diminishing parameter capacity and cross-task interference. Overall, while parameter isolation mitigates forgetting, these methods show limitations in knowledge transfer efficiency and long-sequence adaptability.

(3) Degradation of Regularization-based Methods. Regularization methods (EWC, MAS, LwF, VCL) perform poorly on OCW20, with average scores below 0.25. The performance curves for EWC and VCL plateau in the latter stages, indicating a severely diminished capacity to learn new tasks. Results suggest that while regularization limits parameter drift, the accumulation of complex constraint terms complicates optimization as the number of tasks increases. Interestingly, simple L2 regularization outperforms the more complex EWC and VCL, suggesting that simple global constraints may be more stable than importance-based estimation in long-sequence scenarios.

(4) Limitations of Experience Replay Methods. Perfect Memory (PM) and A-GEM perform poorly on OCW20. PM is limited by buffer capacity relative to the total data volume, while A-GEM's gradient projection constraints appear overly restrictive, hindering new task learning. The suboptimal performance of these methods indicates that simple data replay is insufficient to effectively address catastrophic forgetting in this long-sequence continual learning setting.

### F.2. FWT Decomposition on OCW20

Since OCW20 repeats the OCW10 task sequence, the overall FWT combines two effects: transfer to previously unseen tasks during the first pass and activation of previously learned knowledge during the second pass. We therefore decompose the OCW20 FWT in Table 9.

As shown in Table 9, the decomposition shows modest but positive transfer to unseen tasks in the first half, which is expected because OCW10 contains diverse manipulation primitives. In the second half, HTAC achieves strong knowledge activation on revisited tasks, indicating that semantic routing can activate relevant frozen experts and reuse previously acquired knowledge.

*Table 9.* FWT decomposition on OCW20.

| Component | FWT |
| --- | --- |
| Tasks 2–10: novel-task transfer | $0.018_{\pm 0.019}$ |
| Tasks 11–20: knowledge activation | $0.700_{\pm 0.016}$ |
| Overall OCW20 FWT | $\mathbf{0.380_{\pm 0.020}}$ |

## F.3. Multi-Perspective Analysis of Long-Sequence Learning

To further elucidate the advantages of HTAC in long-sequence tasks, we leverage the performance matrix heatmap (Figure 6) and forward transfer metrics to analyze the method from three perspectives: knowledge retention and reuse, forward transfer capability, and task-specific performance.

Figure 6 presents the performance matrix heatmaps for all methods on the OCW20 benchmark. In these matrices, rows correspond to training stages (i.e., after completing task $i$), while columns denote test tasks (task $j$). The region enclosed by the red dashed triangle highlights the lower triangular area, indicating the retention of previously learned tasks, whereas the green dashed diagonal denotes the performance on the current task immediately after training.

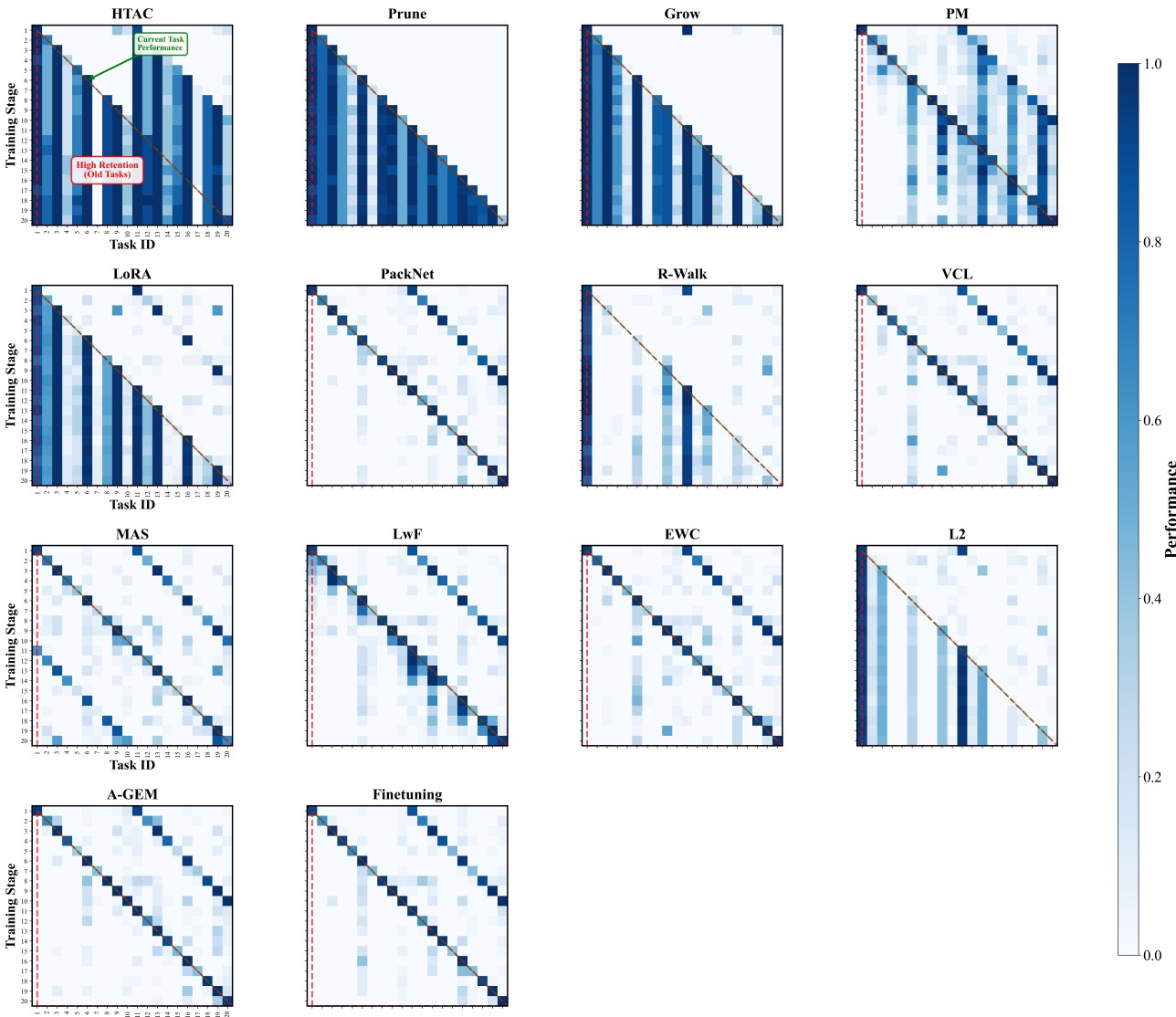

*Figure 6.* Performance matrices for all 14 methods on OCW20. Each cell (i,j) shows the performance on test task j after training on task i. The red dashed triangle highlights the retention area, and the green diagonal represents current task performance. All results are averaged over 3 random seeds.

(1) Knowledge Reuse and Sharing in HTAC. The HTAC heatmap exhibits a distinct dark blue lower triangle, indicating sustained high performance on historical tasks following new task acquisition. Crucially, the intensity of the retention area remains stable throughout the 20-task sequence, showing no significant decay. This stability is directly attributable to HTAC's parameter freezing mechanism: task-specific experts and adapters are frozen post-training, preventing the

overwriting of historical knowledge. Furthermore, the dark blue diagonal indicates efficient utilization of compositional knowledge, with current task performance approaching the single-task upper bound. Notably, starting from task 11, the matrix displays a block-diagonal pattern, suggesting that revisited tasks not only activate their specific historical experts but also benefit from shared knowledge within the same domain, validating the effectiveness of the domain-level expert sharing mechanism.

(2) Knowledge Reuse in Baselines. Baseline methods exhibit distinct retention characteristics. Finetuning shows a sharp contrast with a white lower triangle and dark blue diagonal, reflecting catastrophic forgetting where new learning overwrites old knowledge. Prune and Grow display a light blue lower triangle (average retention 0.4-0.6), indicating partial retention due to shared parameters between independent sub-networks being updated. Regularization methods (MAS, EWC) show a "gradual forgetting pattern," where retention fades from recent to distant tasks, highlighting the limitation of importance-based methods in capturing the long-term value of parameters. LoRA presents a uniform medium blue heatmap, reflecting the "rank bottleneck" where fixed-rank constraints limit performance across all tasks equally.

(3) Differences in Forward Transfer Capability. The task revisitation in OCW20 allows for quantitative analysis of Forward Transfer (FWT). Table 1 shows HTAC achieves significant FWT improvement, driven by two mechanisms: First, for tasks 11-20, HTAC uses SBERT to identify semantic correspondence with tasks 1-10, enabling the composition module to instantly reactivate the relevant task and domain experts for zero-shot transfer. This is visually confirmed by the matching intensity of the 11th and 1st rows in the heatmap. Second, for tasks within the same domain, the domain expert already encodes accumulated domain-level knowledge, reducing the new task to learning only task-specific differences. In contrast, baselines show low or negative FWT. Prune and Grow are limited by output-level composition, preventing fine-grained modular reuse. Regularization methods suffer from negative transfer due to gradient interference, while LoRA lacks explicit knowledge sharing, requiring retraining for revisited tasks.

(4) Task-Specific Performance. Heatmap column analysis reveals task-specific strengths. For Hammer-v2 (Task 1, 11), HTAC maintains high retention at the end of the sequence, whereas Finetuning drops to near zero. Notably, HTAC's performance on the revisited Hammer task (Task 11) exceeds its initial performance (Task 1), demonstrating positive transfer from manipulation skills learned in the intervening tasks. For the complex Stick-pull-v2 (Task 5,15), where baselines struggle, HTAC achieves high performance that further improves upon revisitation, confirming that knowledge accumulation mitigates learning difficulty. In the Pushing domain (Task 7,17), HTAC shows cross-task retention where learning Push-v2 sustains performance on related tasks like push-wall, providing direct evidence of the domain expert sharing mechanism.

In summary, our comprehensive analysis on the OCW20 benchmark strongly validates the superiority of HTAC's core mechanisms—hierarchical knowledge representation, semantic-based composition, and parameter isolation—in long-sequence continual learning scenarios, highlighting its significant advantages in knowledge retention and forward transfer.

