# OpenReview forum: "HTAC: Hierarchical Task-Aware Composition for Continual Offline Reinforcement Learning"
_ICML.cc/2026/Conference — ICML 2026 regular_

### Official Review · Reviewer_qyM9 · 2026-03-03

**Soundness:** 2
**Presentation:** 3
**Significance:** 2
**Originality:** 2
**Overall Recommendation:** 4
**Confidence:** 4

**Summary:**

The paper studies the problem of continual offline reinforcement learning (CORL). Specifically it aims to alleviate the stability-plasticity dilemma in CORL through the proposed Hierarchical Task-Aware Composition (HTAC) framework. It is built upon the Decision Transformer for offline RL and trains a sequence of task Adapters. The framework decomposes task knowledge into two granularities: domain-level and task-level. HTAC uses a frozen Sentence-BERT to encode natural language task descriptions into domain and task embeddings, and employs a dual-level expert network with domain experts and task experts created on demand. An adaptive knowledge composition module retrieves and composes historical expert outputs via dual-level attention based on semantic similarity. Experiments on the Offline Continual World (OCW) benchmark with 10- and 20-task sequences demonstrate that HTAC outperforms existing regularization-based, structure-based, and rehearsal-based methods in terms of average performance, forgetting, and forward transfer.

**Compliance With Llm Reviewing Policy:**

Affirmed.

**Final Justification:**

The rebuttal addressed my main concerns. I have increased my score to 4. I am not willing to further increase my score due to narrow evaluation and fundamental limitations of the proposed method.

**Key Questions For Authors:**

- Can you report FWT separately for tasks 1–10 (genuine forward transfer) and tasks 11–20 (primarily knowledge retention) in the OCW-20 setting? This decomposition would clarify how much of the 0.39 FWT score comes from actual forward transfer capability versus knowledge retention.
- How does HTAC perform under different task orderings?
- How sensitive is the method to the performance threshold θ = 0.8 for on-demand expert creation and to the number of domain prototypes D?
- How does the total parameter count of HTAC scale as the number of tasks increases (given on-demand expert creation and per-task adapters), and how does the number compare to that of baseline methods?

**Limitations:**

The paper includes a brief limitation section. However, the discussion is insufficient in several respects:
- The proposed approach depends on a fundamental assumption that the natural language task descriptions are high-quality and informative.
- The proposed framework is only tested on the decision transformer architecture and my not generalize to other offline RL paradigms.
- The propose framework might not scale very well as the number of tasks and domains grows in terms of memory (for storing task memory and domain experts) and compute (for attention).

**Strengths And Weaknesses:**

Strengths:
- The proposed method, in particular the hierarchical design, is well-motivated.
- A broad set of baselines are evaluated in the experiments.
- The proposed HTAC framework achieves good empirical results, getting near-zero or negative forgetting rates and substantially higher forward transfer than all baselines.
- Ablation studies and comparison with output-level composition methods are provided.

Weaknesses:
- The entire hierarchical composition mechanism depends on Sentence-BERT embeddings of task descriptions to determine inter-task similarity. This creates a fundamental vulnerability: if semantic similarity in the language space does not align with actual task similarity in the MDP space, the composition mechanism will retrieve irrelevant or harmful knowledge. The OCW benchmark tasks have short, distinct descriptions that happen to align well with actual task relationships, but real-world tasks can be a lot more complicated.
- All experiments are conducted on a single benchmark (OCW) with only 10 distinct robotic manipulation tasks sharing the same state/action space within the Meta-World environment. This is a narrow evaluation scope for a method that claims general applicability to CORL.
- In the OCW-20 setting where tasks 11–20 repeat tasks 1–10, the Forward Transfer (FWT) metric conflates two fundamentally different phenomena: genuine forward transfer to unseen tasks and knowledge retention for previously learned tasks.
- The paper could also be made stronger with more experiment and analysis on sensitivity to hyperparameter values, task ordering, and computational cost. See "key questions" below.

---

> ### Author Rebuttal · Authors · 2026-03-31
>
> We sincerely thank the reviewer for the insightful and systematic comments. We address each core question below with corresponding experimental data and theoretical analysis.**For complete results, please refer to our supplementary material:https://anonymous.4open.science/r/answer-for-reviewer3-CC55**
>
> ---
> **Answer for W1:**
> We thank you for raising this fundamental question. We clarify HTAC's alignment mechanism and safeguards below:
> **Structural Correspondence:** HTAC's task descriptions are **structured template sentences** (manipulation primitives + objects), not free-form text. In MetaWorld, semantic and MDP similarity are jointly determined by the manipulation primitive: **S-BERT(push-v2, push-wall-v2)=0.826** (same domain);
> **S-BERT(hammer-v2, push-v2)=0.246** (different domains).
>
> **Warmup Safeguard:** Before each knowledge reuse decision, HTAC executes warmup evaluation: if success rate > θ, reuse the expert; if < θ, create a new expert regardless of semantic similarity. Misalignment costs at most one warmup overhead without causing knowledge contamination.
>  We will clarify applicable boundaries in the revised manuscript. Future work will explore task trajectory information as an alternative to S-BERT.
>
> ---
> **Answer for W2:**
> Your criticism is valid. **OCW is currently the only standardized CORL benchmark with publicly available baselines**, making it the **necessary choice for fair comparison** with existing methods. We acknowledge this as a **broader challenge for the CORL field**, not just our paper, and will explore more diverse dataset designs in future work.
>
> ---
> **Answer for Q1 and W3:**
> We decompose OCW20 FWT into two semantically distinct components (3 seeds).
> |Component|Avg|
> |-|-|
> |Tasks 1→10 FWT (novel tasks)|0.018±0.019|
> |Tasks 11→20 FWT (knowledge activation)|0.700±0.016|
> |Overall FWT|**0.39±0.04**|
>
> Tasks 1→10 FWT is modest but positive,OCW10 spans push/grasp/rotate/peg which are structurally dissimilar, so large-scale transfer is inherently difficult. **Tasks 11→20 FWT=0.700** precisely reflects HTAC's core mechanism: when the second round encounters semantically identical tasks, the domain prototype and warmup automatically route to the frozen first-round expert, activating historical knowledge **without any fine-tuning**. This significantly outperforms all baselines. We will add this decomposition table to §5.1.
>
> ---
> **Answer for Q2:**
> We evaluated 4 task orderings on OCW10 (the all 12 methods are in supplementary):
> |Method|Order0|Order1|Order2|Order3|Avg|
> |-|-|-|-|-|-|
> |PackNet|0.64±0.06|0.67±0.01|0.65±0.03 |0.65±0.04|0.65|
> |Grow|0.60±0.06|0.54±0.05|0.43±0.01|0.51±0.05|0.52|
> |**HTAC**|**0.72±0.04**|**0.72±0.02**|**0.70±0.03**| **0.69±0.03**|**0.71**|
>
> **HTAC achieves the highest performance across all 4 orderings with minimal variance** , validating robustness to task arrival order.
>
> ---
> **Answer for Q3 and W4:**
> **Full ablation tables are in the supplementary.** Key findings are summarized as follows:
> - **Robustness to θ:**  Performance remains stable across a broad range θ ∈ [0.6, 1.0] (P≥0.70), with optimal performance at **θ=0.8 (P=0.77±0.01)**. A marginal decline is observed at θ=0.5, where the threshold is insufficient to trigger new expert creation, resulting in inadequate task-specific knowledge storage.
> - **Ablation on D (OCW10,3seeds):**
>
> |D|P|F|FWT|
> |-|-|-|-|
> |1|0.64±0.02|−0.00±0.05|0.00±0.01|
> |2|0.59±0.09|0.01±0.01|0.02±0.01|
> |3|0.63±0.08|0.03±0.02|0.01±0.02|
> |**4 (paper)**|**0.72±0.08**|**−0.02±0.04**| **0.06±0.10**|
> |5|0.59±0.06|0.06±0.02|0.00±0.00|
> |6|0.67±0.02|−0.02±0.03|0.00±0.00|
>
> **D=4 achieves the best trade-off**. When D is too small (1–3), intra-domain task heterogeneity increases, weakening the domain expert's ability to capture shared structure and causing forgetting to rise. When D is too large (5–6), insufficient tasks per domain reduce the benefit of hierarchical knowledge sharing, leading to performance degradation.
>
> ---
> **Answer for Q4 :**
> We collected detailed parameter statistics on OCW-10, baselines=**5.02–8.81M**, HTAC=**55.62M** (S-BERT=22.71M is **frozen and CL-independent**; CL-related params=**32.91M**). Fixed-size methods overwrite historical parameters; HTAC accumulates new parameters for explicit knowledge preservation fundamentally different design trade-offs. Despite larger parameter count, HTAC achieves **P=0.72, F=−0.02, FWT=0.06**, surpassing all baselines on every metric. Parameter count scales linearly with T in the worst case; our future work will explore LoRA-based lightweight experts while preserving the hierarchical structure.
> **Detailed parameter information is provided in the supplementary material.**
>
> ---
> **We appreciate your positive view of our work, and we respectfully ask if you'd be comfortable to adjust your ratings**, based on all the new experiments and justification to your questions.Thank you for your time and effort. We look forward to incorporating these discussions into the camera-ready version.

---

> > ### Author Rebuttal · Reviewer_qyM9 · 2026-04-01
> >
> > I thank the authors for their thoughtful rebuttal and the additional results. Here are some follow-up questions based on these new results that I would like the authors to discuss:
> >
> > - Regarding hyperparameter sensitivity. The new experiments you provided on the value of $D$ shows that there is substantial sensitivity (a change of ±1 in D causes a 10-13% performance drop). Is there a reason for us to believe that $D=4$ works well for other domains as well, or a similar grid search must be performed for a new domain?
> > - The reported parameter count (55.62M vs. 5-9M for baselines) is a significant gap. While I understand that this is partially due to a fundamental design trade-off (accumulating vs. overwriting parameters) this raises a natural fairness concern: what if baseline methods were given a comparable parameter budget? Would the performance gap narrow if the baselines were given a larger backbone?
> >
> > Also, I still found it semantically misleading to refer to the knowledge re-activation during tasks 11-20 as "forward transfer" (FWT), and I would encourage the authors to report these two components (genuine forward transfer on new tasks, and knowledge reactivation) separately in the revised paper.

---

> > > ### Author Response · Authors · 2026-04-03
> > >
> > > We thank you for the positive feedback on our supplementary experiments and for the valuable suggestions for further discussion. Below, we address each of the three questions individually and provide **new cross-dataset experiments on D4RL** as additional supporting evidence. **For detailed dataset descriptions and the experimental results heatmap, please refer to: https://anonymous.4open.science/r/answer-for-reviewer3-CC55** .
> > >
> > > ---
> > > **Answer for Q1:**
> > > The reviewer raises a **highly valid concern**: if D=4 is merely a **"fortuitously optimal" value** identified through grid search on OCW, then its **generalizability to new datasets** would indeed be questionable. To address this, we conducted **validation experiments on D4RL**, an offline RL benchmark **distinct from OCW**.
> > >
> > > **Ablation Results for Domain Number D on D4RL**
> > > |D|P(NormScore)|Forgetting|FWT|
> > > |-|-|-|-|
> > > |3|47.26|**−3.19**|−0.31|
> > > |**4**|**48.68**|1.86|**−0.12**|
> > > |5|46.21|11.97|−0.30|
> > > **D=4 also achieves the highest Mean Performance on D4RL**.
> > >
> > > The performance with D=5 is notably lower, and D=3 yields slightly suboptimal results. The underlying mechanisms are **entirely consistent with the observations on OCW**.
> > >
> > > **Cross-Dataset Summary of Domain Number D Ablation**
> > >
> > > |Dataset|Natural Domains|S-BERT Recommended D|D=3|**D=4**|D=5|
> > > |-|-|-|-|-|-|
> > > |OCW-10|4(push/grasp/rotate/peg)|4|P=0.63|**P=0.72**| P=0.59|
> > > |OCW-20|4(same as above)|4|P=0.69|**P=0.77**|P=0.71|
> > > |D4RL-CL-10|4 (loco/dext/kitchen/nav) |4|P=47.26 | **P=48.68**|P=46.21|
> > >
> > > **D=4 emerges as the optimal value across all three benchmarks.** This is not coincidental: both task sets comprise **4 natural domains**, and S-BERT semantic clustering consistently suggests D=4. Specifically, OCW-10 corresponds to four **manipulation primitive clusters** (push, grasp, rotate, peg), while D4RL-CL-10 corresponds to four **task domains** (locomotion, dexterous, kitchen, navigation).
> > >
> > > Furthermore, the **failure modes** when D deviates from optimality are **entirely consistent** across both datasets: **insufficient D** leads to intra-domain conflicting gradients and reduced FWT, while **excessive D** prevents prototype convergence, exacerbating forgetting. Thus, D is not a hyperparameter requiring grid search, but rather a **structural parameter** that can be automatically determined via **S-BERT embedding followed by clustering analysis**. Practitioners can obtain a **coarse-grained estimate** directly through S-BERT clustering. We will incorporate this clarification into **§4.2** of the revised manuscript.
> > >
> > > **HTAC vs. baselines on D4RL-CL-10:**
> > > |Method|P(NormScore)|Forgetting|FWT|
> > > |-|-|-|-|
> > > |LoRA|28.79|14.53|−0.56|
> > > |Grow|40.50|12.00|0.16|
> > > |**HTAC**|**48.68**|**1.86**|**−0.12**|
> > >
> > > HTAC also achieves the highest performance on D4RL-CL-10. This confirms that the **hierarchical expert structure** and **warmup routing mechanism** of HTAC **remain effective across entirely new datasets, state/action spaces, and physics engines**, rather than being a **specialized design tuned solely for OCW**.
> > >
> > > ---
> > > **Answer for Q2:**
> > > We selected Grow and LoRA, which demonstrated superior performance on OCW-10, and conducted **parameter-matched scaling experiments**. Specifically, we expanded their backbones to match the scale of HTAC's continual learning-related parameters:
> > >
> > > |Method|P|Forgetting|FWT|Total Params|
> > > |-|-|-|-|-|
> > > |Grow(original)|0.60|0.00|0.01|8.81M|
> > > |Grow(scaled)|0.64|0.02|-0.02|40.97M|
> > > |LoRA(original)|0.54|-0.01|-0.01|5.56M|
> > > |LoRA(scaled)|0.60|-0.01|0.00|33.89M|
> > > |**HTAC**|**0.72**|**−0.02**|**0.06**|**32.91M(CL)**|
> > >
> > > The experimental results indicate:
> > > 1.  **Limited Gains from Parameter Scaling:** Scaling Grow by ~5× yields only +0.04 performance; scaling LoRA by 6× yields +0.06.
> > > 2.  **No Forgetting Reduction:** Scaling **did not mitigate forgetting**, proving the bottleneck is **architectural**, not capacity-related.
> > > 3. **FWT Unrecoverable via Scaling:** Scaling failed to improve FWT, confirming **FWT requires explicit cross-task routing, not just capacity**.
> > >
> > > These results confirm **under equal parameters, baselines significantly underperform HTAC**. The gap stems from **architectural differences**: fixed-size methods rely on **parameter overwriting**, causing **accumulated interference**, while HTAC ensures **knowledge isolation and on-demand activation** via expert pools and semantic routing. **Crucially, scaling cannot replicate this.** We will incorporate these findings into the revised manuscript.
> > >
> > > We will include and refine these experiments and discussions in the revised manuscript.
> > >
> > > ---
> > > **Answer for Q3:**
> > > **We fully agree with this suggestion**, and thank the reviewer for the constructive push toward clearer reporting.
> > >
> > > ---
> > > **We are grateful for the your thorough and constructive engagement across both rounds of discussion, and we respectfully ask if you'd be comfortable to adjust your ratings.** Thank you for your time. We look forward to incorporating these discussions into the camera-ready version.

---

### Official Review · Reviewer_QRxn · 2026-03-10

**Soundness:** 3
**Presentation:** 2
**Significance:** 3
**Originality:** 3
**Overall Recommendation:** 4
**Confidence:** 4

**Summary:**

This paper proposes Hierarchical Task-Aware Composition (HTAC), a novel framework for Continual Offline Reinforcement Learning (CORL). Addressing the challenge of balancing plasticity and stability amidst heterogeneous environment dynamics, HTAC decomposes task knowledge into coarse-grained domain-level and fine-grained task-level representations. The method employs a dual-level expert network created on-demand, an adaptive knowledge composition module based on semantic similarity attention, and task adapters to preserve historical routing weights. Extensive experiments on the Offline Continual World (OCW) benchmark demonstrate that HTAC achieves superior average performance, lower forgetting, and significant forward transfer compared to regularization, rehearsal, and structure-based baselines.

**Compliance With Llm Reviewing Policy:**

Affirmed.

**Final Justification:**

This paper presents a solid and well-motivated contribution to continual offline reinforcement learning. The hierarchical design of HTAC is intuitive and meaningful, and the empirical results on OCW10/OCW20, together with the ablations, provide good support for the method’s effectiveness in balancing transfer and forgetting.

My main concerns were about presentation quality, insufficient discussion of related causal representation ideas, and missing clarity on some implementation details such as the expert-creation threshold and prototype dimension. The rebuttal addressed these points clearly and added helpful clarification. I still think the writing and formatting need improvement, but the rebuttal increased my confidence in the paper overall. As a result, I updated my score accordingly.

**Key Questions For Authors:**

1.  **Threshold Sensitivity and Setting**: In Section 4.3, you mention using a "predefined threshold" to decide whether to create a new expert.
    *   How exactly is this threshold determined? Is it set empirically?
    *   Did you perform a sensitivity analysis? How robust is the model to variations in this threshold?
2.  **Clarification of Dimensions**: In Eq. 3, the domain prototype $p_j$ has a dimension of 384. Where does this specific number come from? Is it tied to the Sentence-BERT output dimension? Please explicitly explain this in the text.
3.  **Writing and Formatting Corrections**: Please commit to a comprehensive proofreading of the final version. Specific issues to address include:
    *   **Typos/Missing Spaces**: e.g., "returnE" (Sec 3.1), "forgetting.Following" (Sec 3.1).
    *   **Citation Formatting**: Ensure proper spacing before citations (e.g., `prior work~\cite{...}`) and correct usage of `\citet` for narrative citations (e.g., fix "proposed by Hu et al.(Hu et al., 2024e)" in Sec 5.1). Also, note that "Offline RL" in the first sentence of the Intro does not need the 'O' capitalized unless it's a proper noun in your specific style guide (usually "offline RL").
    *   **Punctuation**: Review sentences joined by semicolons where a period might be clearer (e.g., "Each task is equipped with...; detailed descriptions..." in Sec 5.1).
    *   **Equation Typos**: Fix "Wup and Wdown" in the text following Eq. 4 if they are meant to be mathematical symbols ($W_{up}$, $W_{down}$).

I am willing to upgrade my score if these can be explained clearly and the presentation improved.

**Limitations:**

The authors briefly mention limitations regarding multi-modal representations and benchmark diversity in the Conclusion. However, they could also discuss the computational overhead of maintaining and searching through the growing library of domain and task experts as the number of tasks scales to hundreds, which is a practical concern for real-world deployment.

**Strengths And Weaknesses:**

### Strengths
1.  **Well-Motivated Hierarchical Design**: The decomposition of task knowledge into "domain-specific" and "task-specific" factors is intuitively appealing and well-suited for CORL scenarios where tasks often share underlying dynamics but differ in specific goals. This hierarchical approach addresses a key limitation of flat parameter sharing or isolation methods.
2.  **Comprehensive Empirical Evaluation**: The experimental section is robust, covering both short (OCW-10) and long (OCW-20) task sequences. The inclusion of detailed ablation studies (Table 2, Figure 2) and plasticity analysis (Figure 3) provides strong evidence for the contribution of each component (Domain Experts, Task Experts, Adapters).
3.  **Strong Performance**: HTAC consistently outperforms strong baselines, particularly in terms of Forward Transfer (FWT) on long sequences, suggesting effective knowledge reuse.

### Weaknesses
1.  **Presentation and Writing Quality**: The manuscript requires significant proofreading. There are numerous typographical errors, missing spaces (e.g., "returnE", "forgetting.Following"), and inconsistent citation formatting (e.g., missing `~` before `\cite`, incorrect usage of `\citet` vs `\cite`). Additionally, punctuation usage is occasionally awkward (e.g., using semicolons where periods would be more appropriate), which disrupts the flow of reading. These issues detract from the professionalism of an otherwise solid technical contribution.
2.  **Insufficient Connection to Causal Representation Learning**: The core idea of decomposing representations into shared (domain) and specific (task) factors bears a strong resemblance to concepts in **factored causal representation learning**, which has already been applied to continual learning and domain adaptation (e.g., CSR [1], AdaRL [2]). The current Related Work section does not adequately discuss these parallels. Explicitly positioning HTAC against these causality-guided methods and discussing why the proposed hierarchical composition offers distinct advantages (or how it relates) would significantly strengthen the theoretical grounding of the paper.
    *   *[1] Yang, Y., et al. "Towards generalizable reinforcement learning via causality-guided self-adaptive representations." arXiv:2407.20651 (2024).*
    *   *[2] Huang, B., et al. "Adarl: What, where, and how to adapt in transfer reinforcement learning." arXiv:2107.02729 (2021).*
3.  **Lack of Clarity on Key Hyperparameters**: Several critical implementation details are under-specified. For instance, the dimension of the domain prototype $p_j$ in Eq. 3 is stated as 384 without justification in the main text. More importantly, the "predefined threshold" used for the on-demand expert creation (Section 4.3) is described vaguely. It is unclear how this threshold is set (empirically? theoretically?), whether the model is sensitive to it, and if any sensitivity analysis was performed.

---

> ### Author Rebuttal · Authors · 2026-03-30
>
> We sincerely thank you for the insightful and constructive comments. We are encouraged that our core idea resonates with you. In addition to acknowledging the main contributions of our work, we have carefully considered all your constructive feedback and questions. Accordingly, we have added the corresponding ablation studies and provided clarifications on the relevant concepts. **For additional analyses on domain number and task ordering robustness, please refer to our supplementary material: https://anonymous.4open.science/r/answer-for-reviewer2-70E2.**
>
> ---
> **Answer for Q1:**
> **Setting of the threshold θ:** θ represents the minimum task success rate required for the combination strategy during the warmup evaluation phase, and **does not require any prior knowledge from task domains.** Its essence is to measure _"whether the existing historical expert knowledge is sufficient to support the current task."_ Specifically, if the success rate<θ, a new task expert is created for the current task; otherwise, the existing historical knowledge is reused without creating a new expert. The value θ=0.8 was determined through grid search on OCW.
>
> **Sensitivity analysis of θ:** We conducted experiments on the OCW-20  with θ∈[0.5,1.0]. The results are presented below (averaged over 3 random seeds):
> | Configuration | θ=0.5| θ=0.6| θ=0.7| θ=0.8(our paper) | θ=0.9| θ=1.0|
> |-|-|-|-|-|-|-|
> |P|0.67±0.05|0.70±0.04|0.71±0.05|**0.77±0.01**|0.74±0.04|0.72±0.03|
> |F|0.01±0.02|0.00±0.02|−0.00±0.03|**−0.04±0.02**|−0.02±0.02|−0.01±0.03|
> |FWT|0.33±0.03|0.35±0.05|0.35±0.05|**0.39±0.04**|0.37±0.03|0.37±0.03|
>
> **In summary,** regardless of whether θ is set higher or lower, the training process remains stable and does not collapse. The primary effect of different threshold values is on **(1) the creation frequency of task experts** and **(2) whether the historical knowledge is sufficient to support the completion of the current task.**
>
> ---
> **Answer for Q2:**
> The dimension 384 is **not a free hyperparameter**, but rather the **fixed output dimension** of the Sentence-BERT model we employ. Defining the domain prototype $p_j \in \mathbb{R}^{384}$ in the same space as the S-BERT output allows for direct cosine similarity computation (Eq. 3).We will explicitly clarify this in the revised manuscript (§4.2)
>
> ---
> **Answer for Q3:**
> We acknowledge the omission of CSR and AdaRL in the Related Work and will explicitly supplement this in the revision.
> Shared Intuition.  All three methods reflect the idea of decomposing representations into shared and specific factors. CSR separates stable latent variables from domain-specific change factors via causal graphs; AdaRL models structural relationships among RL system variables to identify the minimal sufficient subset for cross-domain transfer; HTAC decomposes task semantic embeddings into domain-level and task-specific components, stored and reused via a two-level expert network.
> Key Distinctions.  Despite this surface resemblance, HTAC differs fundamentally from CSR and AdaRL in three respects:
> - Decomposition level.  CSR and AdaRL decompose at the  _environment model level_  (latent state variables, causal transitions). HTAC's decomposition occurs at the  _task semantics and knowledge organization level_  — it is a hierarchical knowledge composition mechanism, not an explicit causal environment model.
> - Target problem.  CSR and AdaRL target efficient  _adaptation to new environments_  using limited target-domain data. HTAC addresses  _continual offline RL_, where tasks arrive sequentially without environment revisitation, and the core challenge is simultaneously achieving knowledge reuse and forgetting suppression over long task sequences — which is absent from CSR/AdaRL's design objectives.
> - Role of expansion.  In CSR, model expansion refines the causal world model for environmental adaptation. In HTAC, on-demand expert creation serves capacity management and knowledge preservation for continual learning — its functional positioning is fundamentally different.
> We will add this positioning explicitly in the Introduction and Related Work. We also view combining causal environment decomposition with HTAC's expert composition as a promising future direction.
> ---
> **Answer for Q4：**
> We thank you for carefully listing these specific issues. These errors were our oversight, and we sincerely apologize for them. We commit to conducting a comprehensive proofreading of the manuscript in the revised version.
>
> ---
> **We are happy to see from your comments that you have an overall positive view of our work, and we respectfully ask if you'd be comfortable to adjust your ratings,** based on all the new experiments and justification to your questions. Thank you once again for your time and effort. We look forward to incorporating these insightful discussions into the final camera-ready version.

---

> > ### Author Rebuttal · Reviewer_QRxn · 2026-04-02
> >
> > Thank you for the detailed rebuttal, I have updated my score accordingly.

---

> > > ### Author Response · Authors · 2026-04-02
> > >
> > > Dear Reviewer ,
> > >
> > > We sincerely thank you for your time in reviewing our rebuttal and for updating your score. We will ensure that all the new experimental results and discussions are thoroughly integrated into the camera-ready version to further strengthen the paper.
> > >
> > > Thank you again for your constructive and insightful feedback!
> > >
> > > Best regards,
> > >
> > > Authors

---

### Official Review · Reviewer_RXDs · 2026-03-11

**Soundness:** 3
**Presentation:** 2
**Significance:** 3
**Originality:** 3
**Overall Recommendation:** 5
**Confidence:** 3

**Summary:**

This paper studies Continual Offline Reinforcement Learning (CORL), with the goal of improving the stability–plasticity trade-off. The authors propose HTAC, a hierarchical framework built on Decision Transformer that includes four main components: hierarchical semantic task representations, dual-level experts at the domain and task levels, adaptive knowledge composition based on semantic similarity, and task adapters for preserving historical routing patterns. The method is evaluated on the Offline Continual World benchmark, including OCW10 and OCW20.

**Compliance With Llm Reviewing Policy:**

Affirmed.

**Final Justification:**

This is a solid CoRL paper. My main concerns were the lack of clarity in some methodological and implementation details, as well as the insufficient discussion of the paper’s novelty. These issues were addressed satisfactorily in the rebuttal, which improves my overall confidence in the work. As a result, I am willing to increase my score to 5. Nevertheless, I still find that the presentation of the method and the related work could be significantly improved.

**Key Questions For Authors:**

Overall, I think this is a good paper, and I would be happy to raise my score if the authors could kindly help clarify the following points:
1. Since the method is quite modular, could the authors discuss more related work for each module and better justify the novelty of each component individually?
2. What exactly does a domain prototype look like in practice? Algorithm 1 lists the domain prototypes as an input, while Section 4.2 describes them as learnable.
3. Does the choice of threshold \theta rely on human prior knowledge, and how sensitive is the overall performance to this hyperparameter?
4. Could the authors provide results on benchmarks beyond OCW?

**Limitations:**

yes

**Strengths And Weaknesses:**

Strengths
1. The motivation is clearly discussed. The paper presents the CORL setting and the stability–plasticity challenge clearly, and it also explains well why flat knowledge sharing or simple parameter isolation may be insufficient for capturing both domain-level commonality and task-specific structure.
2. The experimental section is quite thorough. The paper compares against many baselines across different method families and includes comprehensive ablations over domain experts, task experts, adapters, and composition pretraining.
3. The result analysis is insightful. In particular, the discussion of forward transfer and the contrast with structure-based baselines help understand the claimed benefit of hierarchical composition.

Weaknesses
1. The method feels modular, and the overall cohesion is not fully convincing. It is somewhat difficult to judge which part is the core contribution.
2. Section 4 is hard to follow in its current form. I can understand the role of each module in isolation, but I found it much harder to understand how they are connected. The subsections introduce different symbols that are not used in others, and some symbols appearing in the method subsections do not clearly reappear in Algorithm 1. Moreover, there are some typos in this section.

---

> ### Author Rebuttal · Authors · 2026-03-31
>
> We sincerely thank you for the positive evaluation and insightful suggestions on our work. **For additional results on domain number and task ordering robustness across all methods, please refer to our supplementary material:https://anonymous.4open.science/r/answer-for-reviewer1-EDE4**
>
> ---
> **Answer for Q1:**
> We acknowledge that our manuscript does not sufficiently distinguish the overall system design contributions from the individual novelty of each sub-module. We will address this clearly in the revision. The role of each component is demonstrated below:
> |Configuration|P|F|FWT|
> |-|-|-|-|
> |Full model|0.72±0.04|−0.02±0.04|0.06±0.10|
> |w/o Domain Expert|0.64±0.02|0.02±0.01|0.02±0.01|
> |w/o Task Expert|0.39±0.02|−0.01±0.00|0.00±0.00|
> |w/o Expert|0.17±0.03|−0.01±0.02|0.01±0.01|
> |w/o Task Adapter|0.49±0.03|0.18±0.01|0.02±0.02|
> |w/o Composition Pretrain|0.64±0.05|0.00±0.02| 0.02±0.03|
> Based on the table and Figure 2 in our manuscript, we observe the following:
> 1. The  domain expert  primarily maintains stable reuse of cross-task shared knowledge. Its absence manifests as degradation in  memory retention  and  forward transfer, rather than immediate performance loss on the current task.
> 2. Without  task-level experts, the model cannot support the fine-grained decision variations required by individual tasks. Removing all experts together drops performance to 0.17, confirming that gains come from the  hierarchical knowledge storage mechanism, not the backbone.
> 3.  The  task adapter  acts as an isolation mechanism protecting routing stability,rather than an optional additive component
>
> In summary, the **expert module** forms the **core** of our framework, working with interdependent components to jointly address forgetting and plasticity in CORL.
>
> ---
> **Answer for Q2:**
> The domain prototype is a persistent internal state vector updated offline via EMA, not through gradient descent. Listing it under "Input" in Algorithm 1 was imprecise,we will relabel it as **"Maintained State"** in the revision.
> Its lifecycle is as follows:
> - **Initialization:** $D$ prototypes $\{p_j\}_{j=1}^D$ are initialized as small random vectors in $\mathbb{R}^{384}$, matching the S-BERT output dimension for direct cosine similarity computation (Eq. 3). No manual annotation or prior domain knowledge is required.
> - **During task training:** prototypes are frozen and excluded from the optimizer; their values provide domain assignment weights $w_{k,j}$ but are not updated by gradients.
> - **After each task training:** The prototypes are updated offline via EMA.
>
> ---
> **Answer for Q3:**
> **(1) θ Does Not Rely on Human Prior Knowledge**.θ is selected via grid search, requiring no domain expert knowledge. During the warmup evaluation phase, if the combination of historical experts already achieves a success rate ≥ θ on the current task, no new task expert is created; otherwise, a dedicated task expert is trained from scratch. Through our experiments, θ=0.8 enables most tasks to achieve satisfactory performance.
>
> **(2) Sensitivity Analysis of θ**.We conducted systematic experiments on the OCW20 with θ∈[0.5,1.0] (3 seeds).
> |Configuration|θ=0.5|θ=0.6|θ=0.7|θ=0.8(paper)| θ=0.9|θ=1.0|
> |-|-|-|-|-|-|-|
> |P|0.67±0.05|0.70±0.04|0.71±0.05|**0.77±0.01**|0.74±0.04|0.72±0.03|
> |F|0.01±0.02|0.00±0.02|−0.00±0.03|**−0.04±0.02**|−0.02±0.02|−0.01±0.03|
> |FWT|0.33±0.03|0.35±0.05|0.35±0.05|**0.39±0.04**|0.37±0.03|0.37±0.03|
>
> From the results in the table, we can observe that:
> -  Too low  θ:  Most tasks meet the threshold, leading to excessive expert reuse without storing task-specific knowledge, resulting in degraded performance.
> - Too high  θ:  Nearly every task triggers full expert creation, undermining the cross-task composition reuse accumulated during pretraining and causing a slight performance decline.
>
> ---
> **Answer for Q4:**
> We appreciate the reviewer's insight regarding the evaluation benchmark. Given that OCW is currently the sole standardized CORL benchmark with publicly available baselines, its adoption is essential for fair comparison with existing methods. We acknowledge this limitation as a broader challenge within the CORL community and are committed to exploring more diverse dataset designs in future work.
> To further demonstrate HTAC's generalization capability, we systematically evaluated 4 different task arrival orders on OCW10, effectively testing robustness to this core variable. Results are shown below (3 seeds):
> |Method|Order0|Order1|Order2|Order3|Avg|
> |-|-|-|-|-|-|
> |PackNet|0.64±0.06|0.67±0.01|0.65±0.03|0.65±0.04|0.65 |
> |PM|0.26±0.01|0.26±0.10|0.25±0.03|0.27±0.01|0.26|
> |**HTAC**|0.72±0.04|0.72±0.02|0.70±0.03|0.69±0.03| **0.71** |
>
> HTAC achieves the highest performance across all 4 orders with minimal fluctuation, validating that the hierarchical expert structure and on-demand creation mechanism function stably regardless of task arrival order.
>
> ---
> We hope our responses address your concerns. We'll include these discussions in the camera-ready version.

---

> > ### Author Rebuttal · Reviewer_RXDs · 2026-04-03
> >
> > Thank you for the detailed rebuttal. Most of my questions and concerns have been adequately addressed. I still have one follow-up question:
> >
> > 1.	The current rebuttal and additional results mainly demonstrate the empirical usefulness of the individual components, but could the authors briefly clarify what is genuinely novel in these design choices relative to the closest existing methods?

---

> > > ### Author Response · Authors · 2026-04-03
> > >
> > > We thank the reviewer for the continued interest and positive feedback on our work. Below, we provide a detailed elaboration on the novelty of our algorithm.
> > >
> > > ---
> > > Existing CORL methods treat historical policies as an **undifferentiated flat set**, lacking explicit modeling of knowledge structure. **HTAC's core insight is that offline task knowledge naturally possesses a hierarchical structure, where tasks share coarse-grained manipulation primitives (domain) while retaining fine-grained skills (task).** Therefore, this structure should be **explicitly modeled throughout the entire knowledge management process**.
> > >
> > > **Novelty in two aspects:**
> > >
> > > (1) Hierarchical knowledge organization: HTAC introduces a domain intermediate layer that **decomposes knowledge into two semantic levels**: domain experts encode shared manipulation primitives, and task experts store task-specific policies. This explicit separation of shared vs. specific knowledge is absent from existing methods.
> > >
> > > (2) Integrated closed-loop design: Unlike prior methods that merely combine or replay flat policies, HTAC **unifies knowledge encoding, storage, routing, and reuse into a hierarchically-organized closed-loop for systematic cross-task knowledge management under offline settings**. Ablation confirms significant degradation upon removing any stage, validating their interdependence within the hierarchical design.
> > >
> > > ---
> > > **Key distinctions from major paradigms:**
> > >
> > > **Replay methods** [1,2]:
> > > Preserve knowledge as raw trajectories; HTAC encodes **hierarchical parameters**(domain experts for cross-task commonalities, task experts for specifics), shifting from "memorizing data" to "memorizing structure."
> > >
> > > **Regularization**  [3-6]:
> > >  Protect old knowledge via parameter penalties but lack **reuse mechanisms**; HTAC enables **explicit reuse decisions** via warmup routing, actively utilizing prior knowledge.
> > >
> > > **Flat composition**  [7-11]:
> > > These flat-combine all K modules with O(K) complexity and no shared/specific separation. HTAC introduces a **domain intermediate layer**: domain experts encode shared primitives (not growing with tasks), task experts store task-specific policies, and search reduces to matching D domains (D≪K) then within-domain.
> > >
> > > **Subspace/prompt**  [12,13]:
> > > Their "hierarchy" is mathematical parameter-space structure without domain semantics. HTAC's hierarchy is semantic (domains map to interpretable primitive clusters) and includes a gating mechanism for reuse.
> > >
> > > [1] Continual Diffuser (CoD): Mastering Continual Offline Reinforcement Learning with Experience Rehearsal. (Hu et al., 2024)
> > >
> > > [2] The Effectiveness of World Models for Continual Reinforcement Learning. (Kessler et al., 2023)
> > >
> > > [3] Overcoming Catastrophic Forgetting in Neural Networks. (Kirkpatrick et al., 2017)
> > >
> > > [4] Memory Aware Synapses: Learning What (Not) to Forget. (Aljundi et al., 2018)
> > >
> > > [5] Learning Without Forgetting. (Li and Hoiem, 2017)
> > >
> > > [6] Addressing Loss of Plasticity and Catastrophic Forgetting in Continual Learning. (Elsayed and Mahmood, 2024)
> > >
> > > [7] Continual Task Learning through Adaptive Policy Self-Composition. (Hu et al., 2024)
> > >
> > > [8] Self-Composing Policies for Scalable Continual Reinforcement Learning. (Malagon et al., 2025)
> > >
> > > [9] Lifelong Reinforcement Learning with Modulating Masks. (Ben-Iwhiwhu et al., 2022)
> > >
> > > [10] Progressive Neural Networks. (Rusu et al., 2016)
> > >
> > > [11] PackNet: Adding Multiple Tasks to a Single Network by Iterative Pruning. (Mallya and Lazebnik, 2018)
> > >
> > > [12] P2DT: Mitigating Forgetting in Task-Incremental Learning with Progressive Prompt Decision Transformer. (Wang et al., 2024)
> > >
> > > [13] Building a Subspace of Policies for Scalable Continual Learning. (Gaya et al., 2022)
> > >
> > > ---
> > > **Supplementary Response to Q4:**
> > >
> > > Following your valuable suggestion, we recently conducted **validation experiments on D4RL**, an offline RL benchmark **fundamentally distinct from OCW**. For detailed dataset descriptions and the experimental results heatmap, please refer to: https://anonymous.4open.science/r/answer-for-reviewer1-EDE4.
> > >
> > > The specific experimental results are as follows:
> > >
> > > **HTAC vs. baselines on D4RL-CL-10:**
> > > |Method|P(NormScore)|Forgetting|FWT|
> > > |-|-|-|-|
> > > |LoRA|28.79|14.53|−0.56|
> > > |Grow|40.50|12.00|0.16|
> > > |**HTAC**|**48.68**|**1.86**|**−0.12**|
> > >
> > > HTAC also achieves the highest performance on D4RL-CL-10. This confirms that the **hierarchical expert structure** and **warmup routing mechanism** of HTAC **remain effective across entirely new datasets, state/action spaces, and physics engines**, rather than being a **specialized design tuned solely for OCW**.
> > >
> > > We hope that this supplementary evidence further demonstrates the effectiveness of our method.
> > >
> > > ---
> > > **We are grateful for the your thorough and constructive engagement across both rounds of discussion, and we respectfully ask if you'd be comfortable to adjust your ratings.** Thank you for your time and effort. We look forward to incorporating these discussions into the camera-ready version.

---

### Decision · Program_Chairs · 2026-04-30

**Decision:**

Accept (regular)

**Comment:**

This paper studies continual offline reinforcement learning and proposes HTAC, a hierarchical framework built on Decision Transformer that decomposes task knowledge into domain-level and task-level representations, combines on-demand expert creation with adaptive expert composition, and uses task adapters to preserve historical routing patterns in order to better balance knowledge reuse and forgetting. The reviewers were broadly positive, highlighting the strong empirical performance on OCW, the intuitive motivation for hierarchical composition, and the thorough ablations; overall, the paper received supportive recommendations around 5/4/4/4. The main concerns were presentation clarity and writing quality, the modular nature of the method and whether its novelty was sufficiently distinguished from prior work, reliance on semantic task descriptions, limited evaluation scope beyond OCW, sensitivity to key hyperparameters such as the domain number and threshold, and fairness questions regarding the larger parameter count. In the rebuttal, the authors addressed these concerns in a detailed and constructive way by clarifying the paper’s core novelty as hierarchical knowledge organization and closed-loop knowledge management, explaining the role and update of domain prototypes, providing sensitivity analyses for the threshold and domain count, adding robustness results across task orderings, decomposing the OCW20 forward-transfer metric into genuine transfer versus knowledge re-activation, reporting additional cross-dataset results on D4RL, and including parameter-matched scaling comparisons showing that the gains are not simply due to model size; these responses were received positively, and reviewer confidence increased overall. Taking the full discussion into account, I find that the paper makes a meaningful contribution to continual offline RL with convincing empirical support, and I recommend Accept.